# Effects of clozapine-N-oxide and compound 21 on sleep in laboratory mice

Janine Traut[1,2,3†], Jose Prius Mengual[2,3,4†], Elise J Meijer[2,3,4], Laura E McKillop[2,3], Hannah Alfonsa[5], Anna Hoerder-Suabedissen[2], Seo Ho Song[6], Kristoffer D Fehér[7,8], Dieter Riemann[1,3], Zoltan Molnar[2], Colin J Akerman[5], Vladyslav V Vyazovskiy[2,3,4], Lukas B Krone[2,3,4,8,9]*

[1]Department of Psychiatry and Psychotherapy, Medical Center - University of Freiburg, Faculty of Medicine, University of Freiburg, Freiburg, Germany; [2]Department of Physiology, Anatomy and Genetics, University of Oxford, Oxford, United Kingdom; [3]Sir Jules Thorn Sleep and Circadian Neuroscience Institute, University of Oxford, Oxford, United Kingdom; [4]The Kavli Institute for Nanoscience Discovery, Oxford, United Kingdom; [5]Department of Pharmacology, University of Oxford, Oxford, United Kingdom; [6]Department of Psychiatry, Beth Israel Deaconess Medical Center, Harvard Medical School, Boston, United States; [7]Geneva University Hospitals (HUG), Division of Psychiatric Specialties, Geneva, Switzerland; [8]University Hospital of Psychiatry and Psychotherapy, University of Bern, Bern, Switzerland; [9]Centre for Experimental Neurology, University of Bern, Bern, Switzerland

*For correspondence:
lukas.krone@dpag.ox.ac.uk

†These authors contributed equally to this work

Competing interest: The authors declare that no competing interests exist.

**Abstract** Designer receptors exclusively activated by designer drugs (DREADDs) are chemogenetic tools for remote control of targeted cell populations using chemical actuators that bind to modified receptors. Despite the popularity of DREADDs in neuroscience and sleep research, potential effects of the DREADD actuator clozapine-N-oxide (CNO) on sleep have never been systematically tested. Here, we show that intraperitoneal injections of commonly used CNO doses (1, 5, and 10 mg/kg) alter sleep in wild-type male laboratory mice. Using electroencephalography (EEG) and electromyography (EMG) to analyse sleep, we found a dose-dependent suppression of rapid eye movement (REM) sleep, changes in EEG spectral power during non-REM (NREM) sleep, and altered sleep architecture in a pattern previously reported for clozapine. Effects of CNO on sleep could arise from back-metabolism to clozapine or binding to endogenous neurotransmitter receptors. Interestingly, we found that the novel DREADD actuator, compound 21 (C21, 3 mg/kg), similarly modulates sleep despite a lack of back-metabolism to clozapine. Our results demonstrate that both CNO and C21 can modulate sleep of mice not expressing DREADD receptors. This implies that back-metabolism to clozapine is not the sole mechanism underlying side effects of chemogenetic actuators. Therefore, any chemogenetic experiment should include a DREADD-free control group injected with the same CNO, C21, or newly developed actuator. We suggest that electrophysiological sleep assessment could serve as a sensitive tool to test the biological inertness of novel chemogenetic actuators.

## Editor's evaluation

This study uses EMG/EEG recordings to demonstrate that the DREADD actuators CNO and C21 have sleep modulatory effects that might result from off-target binding to endogenous receptors. This study highlights the need to use non-DREADD-expressing controls, even when novel actuators are used that cannot convert to clozapine.

**eLife digest** Scientists have developed ways to remotely turn on and off populations of neurons in the brain to test the role they play in behaviour. One technique that is frequently used is chemogenetics. In this approach, specific neurons are genetically modified to contain a special 'designer receptor' which switches cells on or off when its corresponding 'designer drug' is present.

Recent studies have shown that the drug most commonly used in these experiments, clozapine-N-oxide (CNO), is broken down into small amounts of clozapine, an antipsychotic drug that binds to many natural receptors in the brain and modulates sleep. Nevertheless, CNO is still widely believed to not affect animals' sleep-wake patterns which in turn could influence a range of other brain activities and behaviours. However, there have been reports of animals lacking designer receptors still displaying unusual behaviours when administered CNO. This suggests that the breakdown of CNO to clozapine may cause off-target effects which could be skewing the results of chemogenetic studies.

To investigate this possibility, Traut, Mengual et al. treated laboratory mice that do not have a designer receptor with three doses of CNO, and one dose of a new designer drug called compound-21 (C21) that is not broken down to clozapine. They found that high and medium doses of CNO, but also C21 altered the sleep-wake patterns of the mice and their brain activity during sleep. These findings show that CNO and C21 both have sleep-modulating effects on the brain and suggest that these effects are not only due to the production of clozapine, but the drugs binding to off-target natural receptors.

To counteract this, Traut, Mengual et al. recommend optimizing the dose of drugs given to mice, and repeating the experiment on a control group which do not have the designer receptor. This will allow researchers to determine which behavioural changes are the result of turning on or off the neuron population of interest, and which are artefacts caused by the drug itself. They also suggest testing how newly developed designer drugs impact sleep before using them in behavioural experiments. Refining chemogenetic studies in these ways may yield more reliable insights about the role specific groups of cells have in the brain.

## Introduction

Chemogenetics is an important and widely used experimental approach in sleep research (*Weber and Dan, 2016*; *Varin and Bonnavion, 2019*) and could serve as a novel therapeutic strategy in sleep medicine (*Venner et al., 2019*). Designer receptors exclusively activated by designer drugs (DREADDs) enable non-invasive and cell type-specific remote control of neuronal activity in freely moving animals on a time scale of minutes to hours (*Roth, 2016*). In the context of sleep research, these characteristics make DREADD technology a powerful tool with which to probe the contribution of selected neuronal populations in controlling vigilance states (*Hayashi et al., 2015*; *Varin et al., 2018*; *Yu et al., 2019*; *Mondino et al., 2021*), sleep state-specific network oscillations (*Funk et al., 2017*; *Vaidyanathan et al., 2021*), as well as sleep-related physiology (*Harding et al., 2018*; *Fleury Curado et al., 2018*) and behaviour (*Eban-Rothschild et al., 2016*; *Tossell et al., 2020*).

Typically, intraperitoneal injections of clozapine-N-oxide (CNO) are used to activate excitatory hM3Dq or inhibitory hM4Di DREADDs (*Campbell and Marchant, 2018*). Early work suggested that CNO is pharmacologically inert (*Armbruster et al., 2007*) and not back-metabolised to its parent drug clozapine in mice (*Guettier et al., 2009*). However, more recent studies demonstrated relevant conversion of CNO to pharmacologically active metabolites including clozapine (*Gomez et al., 2017*; *Manvich et al., 2018*; *Jendryka et al., 2019*). Clozapine is an atypical antipsychotic drug used in the treatment of schizophrenia with a high affinity for dopamine D2 and serotonin 5-HT$_{2A}$ receptors coupled with a broad binding profile to cholinergic, adrenergic, histaminergic, and serotonergic receptors (*Wenthur and Lindsley, 2013*), which may account for its high efficacy compared to other antipsychotics (*Kane et al., 1988*). In addition, CNO itself was found to present off-target binding at a broad range of neurotransmitter receptors (*Gomez et al., 2017*; *Jendryka et al., 2019*) and to elicit behavioural effects (*Gomez et al., 2017*; *Manvich et al., 2018*; *MacLaren et al., 2016*) at doses commonly used for DREADD experiments.

It is widely thought that CNO does not affect sleep (*Harding et al., 2018*; *Mondino et al., 2021*; *Naganuma et al., 2018*; *Erickson et al., 2019*) and many chemogenetic sleep studies include

convincing control experiments with DREADD-free animals that demonstrate the absence of relevant effects of the chosen CNO preparations and doses on the assessed sleep parameters (*Erickson et al., 2019*; *Mondino et al., 2021*; *Takata et al., 2018*; *Yu et al., 2019*; *Anaclet et al., 2015*; *Venner et al., 2016*). However, a comprehensive assessment of putative dose-dependent effects of CNO on sleep in wild-type mice has never been systematically conducted, although a recent study reported that high CNO doses affected sleep in DREADD-free control animals (*Varin et al., 2018*). This is an important omission, given that clozapine is a sedating antipsychotic drug (*Leucht et al., 2013*) known to modulate sleep in humans (*Hinze-Selch et al., 1997*; *Monti et al., 2017*; *Riemann and Nissen, 2012*) and laboratory rodents (*Spierings et al., 1977*; *Sorge et al., 2004*; *Grønli et al., 2016*; *Coward et al., 1989*). Many of the endogenous neurotransmitter receptors, which are drug targets of clozapine (*Wenthur and Lindsley, 2013*) and to which CNO presents off-target binding affinity (*Jendryka et al., 2019*; *Gomez et al., 2017*), are also involved in the regulation of arousal and sleep (*Saper and Fuller, 2017*).

Initially intended as a control experiment, we here tested whether commonly used doses of CNO (1, 5, and 10 mg/kg) and the novel DREADD agonist compound 21 (C21), which does not convert to clozapine (*Thompson et al., 2018*) but has an off-target binding profile similar to CNO (*Jendryka et al., 2019*), affect sleep in wild-type C57BL/6J mice under laboratory conditions. We find dose-dependent clozapine-like effects of CNO on the proportion of rapid eye movement (REM) sleep, sleep architecture parameters, and frontal EEG power spectra of non-REM (NREM) sleep. In addition, we observed a similar pattern of sleep modulation after injections of a 3 mg/kg dose of C21 resulting in effect sizes comparable to those of the 5 mg/kg CNO condition.

## Results
### CNO suppresses REM sleep
We first assessed the proportion of time spent in wakefulness, NREM, and REM sleep at the beginning of the light period following injections of CNO or saline at light onset (*Figure 1*). We initially focussed on an acute (first 2 hr) time window because CNO concentrations in blood plasma, cerebrospinal fluid, and brain tissue of mice peak within the first 15–30 min after both intraperitoneal and subcutaneous injections (*Jendryka et al., 2019*; *Manvich et al., 2018*), and behavioural side effects are typically tested within the first 2 hr following drug administration (*MacLaren et al., 2016*; *Manvich et al., 2018*; *Gomez et al., 2017*). There was no significant main effect of the treatment condition on the proportion of time spent awake, in NREM sleep or in REM sleep. However, there was a non-significant trend towards a main effect of the treatment condition for NREM sleep ($F_{(1.911, 23.57)}$=3.054, p=0.0682, *Figure 1b*) and REM sleep ($F_{(2.366, 29.18)}$=2.590, p=0.0839, *Figure 1b*) consistent with a report of a short-lasting increase of NREM sleep and decrease of REM sleep following CNO injections of 5 and 10 mg/kg (*Varin et al., 2018*). As the entry to REM sleep requires the previous occurrence of NREM sleep, we also analysed REM sleep as proportion of total sleep time and found a significant effect of CNO treatment on the amount of REM sleep relative to the total sleep time ($F_{(1.783, 21.99)}$=8.951, p=0.0019, *Figure 1b*), due to a significantly reduced REM/NREM sleep ratio following injections of high CNO doses (*Figure 1—figure supplement 1*).

CNO effects might persist for much longer than 2 hr for example due to the later peak of back-metabolised clozapine (*Raper et al., 2017*). To investigate prolonged effects of CNO, we analysed the entire 6 hr observation time window. Again, we found no significant main effect of the treatment condition on the proportion of time spent awake, in NREM sleep or in REM sleep, but a trend towards a reduction of REM sleep ($F_{(2.350, 28.99)}$=2.959, p=0.0601, *Figure 1b*). Relative to the total sleep time, the proportion of REM sleep was significantly altered ($F_{(1.839, 22.68)}$=7.525, p=0.0038, *Figure 1b*) due to a reduced REM/NREM sleep ratio following medium and high doses of CNO (*Figure 1—figure supplement 1*). Effect size calculations for the post hoc comparisons between the individual CNO conditions and the saline condition indicated medium to large effects of CNO on REM sleep (*Supplementary file 1*).

### CNO alters sleep architecture
Physiological sleep in mammals is typically entered through NREM sleep and characterised by the alternation between NREM and REM sleep episodes. The average timing, duration, and frequency

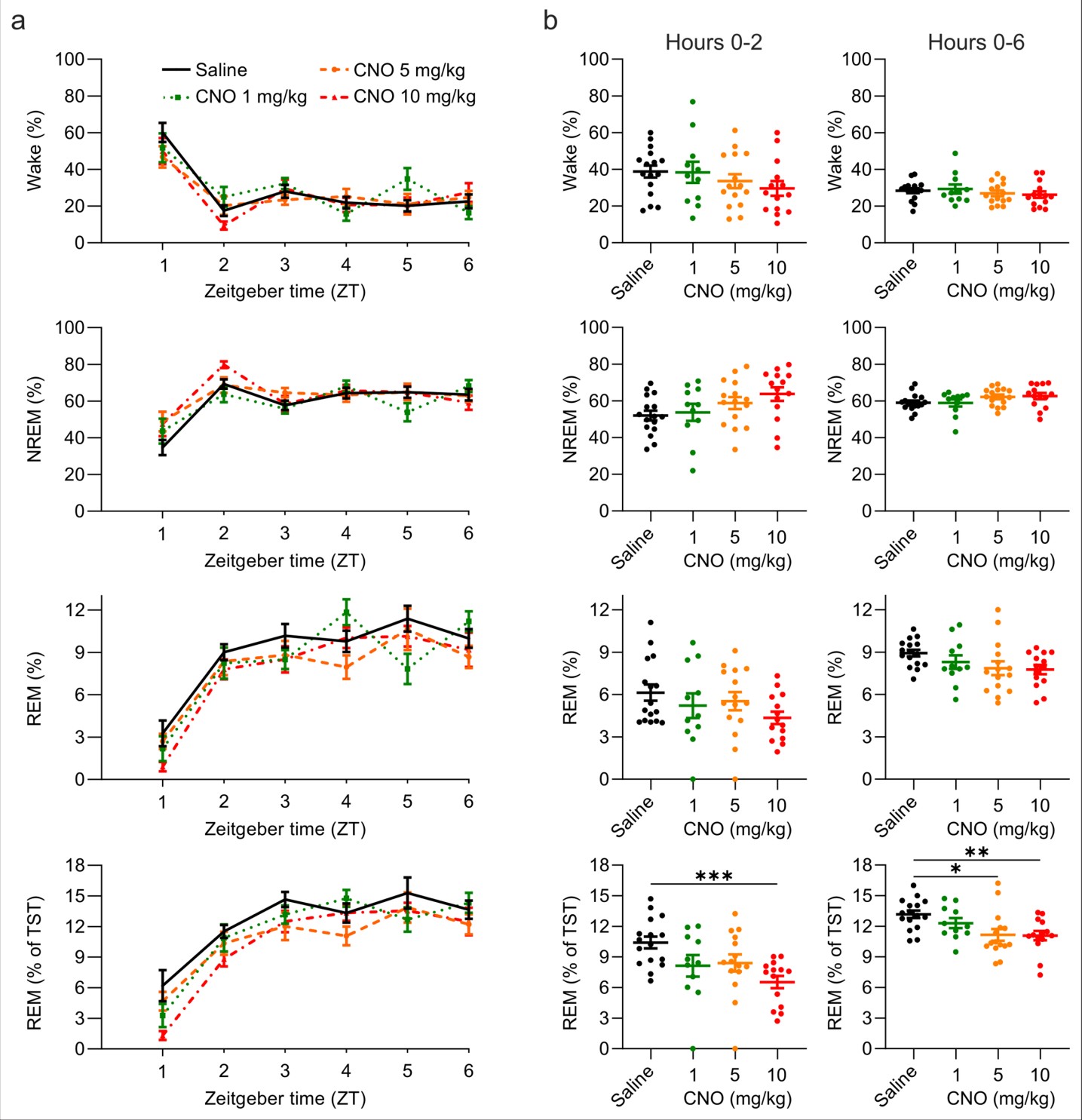

**Figure 1.** Suppression of REM sleep following CNO injection. (**a**) Time course of wakefulness, NREM, and REM sleep in the 6 hr following intraperitoneal injections of CNO or saline at light onset (ZT 0). (**b**) Percentage of time spent in the three vigilance states during the first 2 hr (left column) and over the entire 6 hr observation period (right column) after saline and CNO injections. Note that REM sleep is presented both as proportion of the recording time (third row) and of the total sleep time (fourth row). n=16 for saline, n=11 for 1 mg/kg, n=15 for 5 mg/kg, n=14 for 10 mg/kg. Asterisks indicate post hoc comparisons with significant differences (*p<0.05, **p<0.01, ***p<0.001) for analyses with significant main effects. CNO: clozapine-N-oxide. NREM: non-rapid eye movement sleep. REM: rapid eye movement sleep. TST: total sleep time. ZT: zeitgeber time.

The online version of this article includes the following figure supplement(s) for figure 1:

**Figure supplement 1.** Reduced REM/NREM ratio following CNO injections.

of NREM and REM episodes in mice vary slightly depending on the genetic background but are kept within tight limits for individual strains (*Franken et al., 1999*; *Huber et al., 2000*; *McShane et al., 2010*). The effects of psychotropic drugs on sleep are often most prominent in sleep architecture parameters (*Riemann and Nissen, 2012*). For example, clozapine evokes characteristic changes in sleep architecture in humans (*Monti et al., 2017*), rats (*Sorge et al., 2004*; *Spierings et al., 1977*),

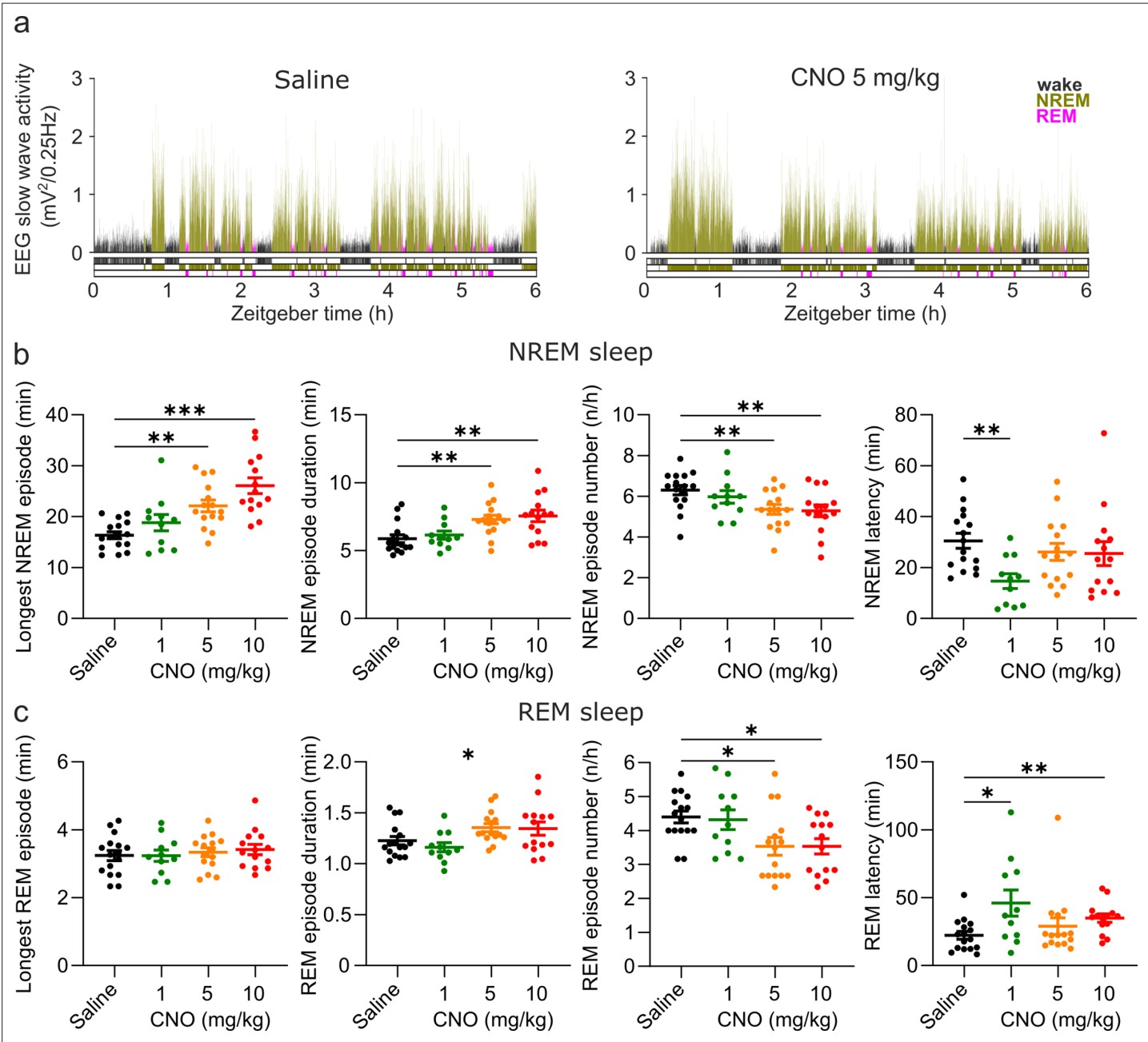

**Figure 2.** Altered sleep architecture following CNO injections. (**a**) Representative hypnograms and EEG slow wave activity (0.5–4.0 Hz, 4 s epochs) from one individual mouse after injection of saline (left panel) and 5 mg/kg CNO (right panel). Note the reduced latency to NREM sleep, the suppression of REM sleep, and the increased duration of individual NREM sleep episodes. (**b**) NREM sleep architecture and (**c**) REM sleep architecture over the 6 hr observation period following saline and CNO injections. Note that for the average REM episode duration there is a main effect of 'treatment condition' but none of the individual post hoc comparisons between CNO and saline reaches the significance level of p=0.05. n=16 for saline, n=11 for 1 mg/kg, n=15 for 5 mg/kg, n=14 for 10 mg/kg for vigilance state analysis in panels b and c. Asterisks indicate post hoc comparisons with significant differences (*p<0.05, **p<0.01, ***p<0.001) for analyses with significant main effects. CNO: clozapine-N-oxide. EEG: electroencephalogram. NREM: non-rapid eye movement sleep. REM: rapid eye movement sleep.

and mice (*Grønli et al., 2016*). Hence, the analysis of sleep architectural parameters is of paramount importance in assessing whether a pharmacological compound modulates sleep.

Characteristics of both NREM and REM episodes were altered by CNO injections (*Figure 2a*). CNO injections elicited unusually long NREM bouts (main effect maximum NREM episode duration: $F_{(2.881,35.53)}$=13.24, p<0.0001). On average, there were longer (main effect mean NREM episode duration: $F_{(2.436,30.04)}$=11.64, p<0.0001) but fewer (main effect NREM episode number: $F_{(2.476,30.54)}$=7.796, p=0.0010) NREM sleep episodes following CNO injections. These effects appeared to be dose-dependent, with higher doses eliciting stronger effects (*Figure 2b*), and were found in both the acute and the prolonged observation period (*Figure 2b*, *Supplementary file 2* and *Supplementary file 3*). The latency between the injection and the onset of NREM sleep was also significantly affected by CNO injections ($F_{(2.146,26.47)}$=3.380, p=0.0463, *Figure 2b*). However, the reduction of the NREM sleep latency was only statistically significant in the 1 mg/kg condition. While the maximum duration of REM sleep episodes was not altered by CNO, there was a significant main effect of the treatment on the mean REM episode duration ($F_{(2.150,26.52)}$=4.230, p=0.0232) and number of REM episodes ($F_{(2.124,26.19)}$=6.430, p=0.0047). Also, the latency between sleep onset and the first transition to REM sleep was significantly changed ($F_{(1.840,22.69)}$=4.691, p=0.0220) with a significantly delayed REM onset in the 1 and 10 mg/kg conditions (*Figure 2b*). Due to the suppression of REM sleep, an analysis of the REM architecture in the acute observation period was not meaningful. In contrast to the effects on sleep states, neither the duration nor the number of wake episodes was modulated by CNO injections (*Supplementary file 2*). In summary, we observed longer but fewer NREM episodes following CNO injections in a dose-dependent fashion, and a similar but less pronounced change in the duration and frequency of REM episodes. Furthermore, CNO injections accelerated sleep onset, but delayed the transition to REM sleep, particularly at low doses.

## CNO affects the NREM spectrogram and sleep consolidation

In addition to sleep time and architecture, EEG spectra are typically analysed in sleep studies. We therefore assessed whether CNO affects EEG spectral power (*Supplementary file 4*). The focus of the EEG spectral analysis was on the comparison between the medium dose (5 mg/kg) of CNO and saline during NREM sleep because these two conditions were counterbalanced and performed first, excluding habituation effects to CNO or to the injection procedure. We observed that CNO injections were followed by a small but significant increase in spectral power in the range between 0.5 and 1.25 Hz and suppression of spectral power in nearly all frequency bins between 6 and 30 Hz during NREM sleep over the first 2 hr (*Figure 3a* and *Figure 3—figure supplement 2*). While the increase in slow frequency bins during NREM sleep appeared to be temporary and did not persist after Benjamini-Hochberg correction for multiple testing (*Benjamini and Hochberg, 1995*), the systematic suppression of power above 6 Hz was more robust and remained significant for both the acute (2 hr) and the prolonged (6 hr) observation period (*Figure 3—figure supplement 2*). Spectral analysis of wakefulness and REM sleep did not reveal any systematic effects of CNO (*Figure 3—figure supplement 2*). A comparison of the other two CNO doses with saline injection indicated that 10 mg/kg CNO elicited similar effects on the NREM sleep spectrogram, but we found no systematic effects of the low CNO dose (1 mg/kg) on EEG spectra of any vigilance state (*Figure 3—figure supplements 1 and 3* and *Supplementary file 4*).

Elevated spectral power in slow frequencies and longer but fewer NREM episodes are typically observed during the initial recovery sleep following sleep deprivation, when sleep is more consolidated (*Huber et al., 2000*). To explore whether the stability of sleep is affected by CNO, we performed a vigilance state transition analysis and assessed the cumulative amount of NREM sleep before the first occurrence of REM sleep, as well as the frequency of brief awakenings (4–16 s intrusions of wake-like EMG and EEG during sleep), which is a behavioural marker of sleep continuity (*Franken et al., 1991*). The state transition analysis showed that in the medium and high dose (5 and 10 mg/kg CNO) conditions, the probability to maintain in both REM and NREM sleep was increased while the transitions between states were unaffected (*Figure 3b*). No such change was observed for the 1 mg/kg CNO condition. For the sleep stability measures there were significant main effects of the treatment condition (NREM before REM: $F_{(2.031,25.04)}$=5.087, p=0.0137; brief awakenings: $F_{(1.968,23.6)}$=10.38, p=0.0006). Effect size calculations indicated a medium to strong increase in NREM sleep before REM onset and reduction in brief awakenings for all CNO doses (*Supplementary file 2*).

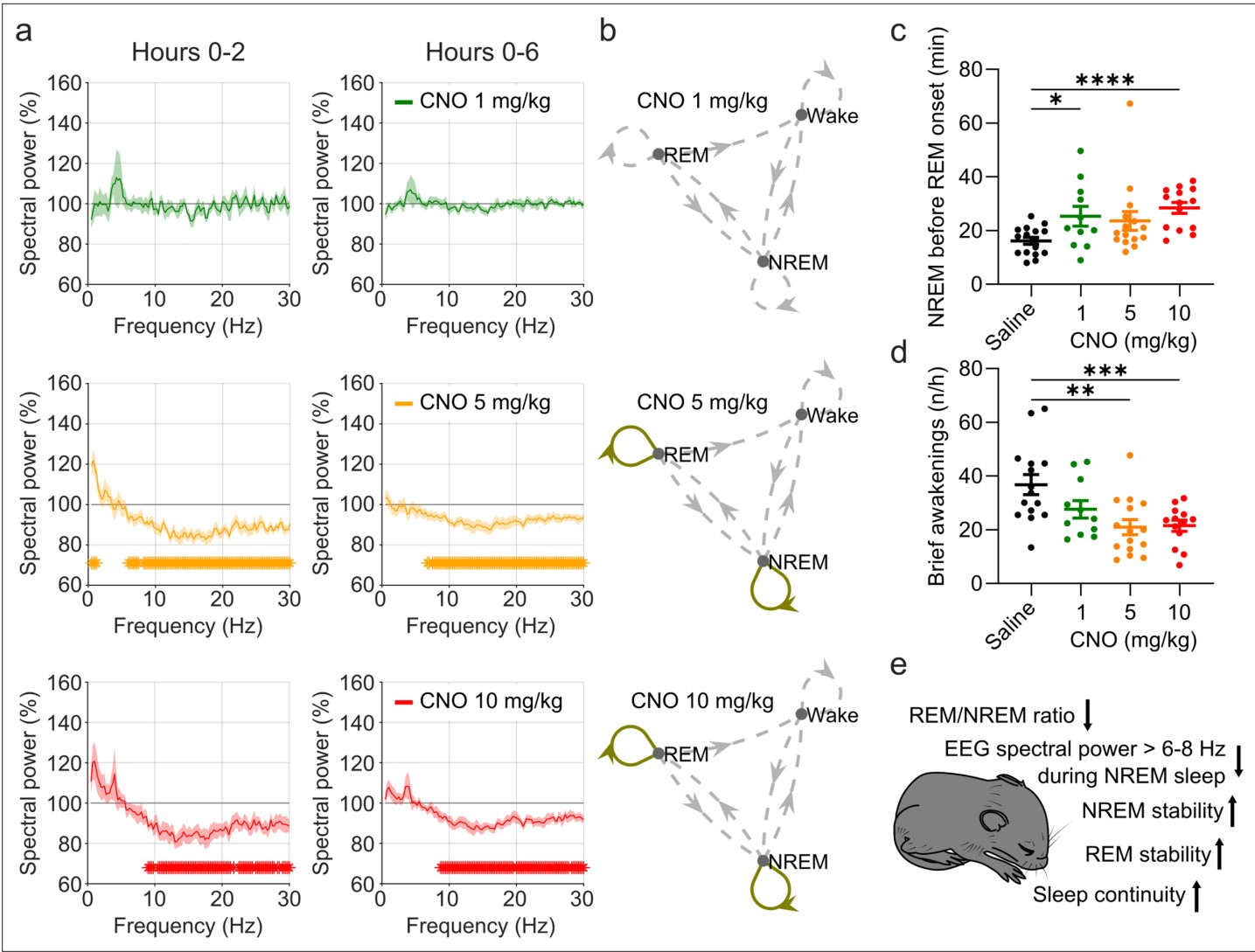

**Figure 3.** EEG spectral changes, increased sleep state stability, and sleep continuity following CNO injections. (**a**) Frontal EEG spectra during NREM sleep following CNO injections relative to saline injections for the acute (first 2 hr, left column) and prolonged (6 hr, right column) observation period. Note the sustained reduction of power in frequency bands >6–8 Hz in the 5 and 10 mg/kg CNO conditions. (**b**) Transitions between vigilance states in the 6 hr period following saline and CNO injections. Note the increased stability of REM and NREM sleep for the 5 mg/kg CNO (REM>REM: p=0.0192, Cohen's d=0.73681; NREM>NREM: p=0.0132, Cohen's d=0.71052) and 10 mg/kg CNO (REM>REM: p=0.0492, Cohen's d=0.65815; NREM>NREM: p=0.0214, Cohen's d=0.77396) condition. Solid olive lines indicate significantly increased transitions/continuations of vigilance states in the respective CNO condition compared to the saline condition, dashed grey lines indicate all possible vigilance state transitions/continuations. (**c**) Cumulative amount of NREM sleep before the first occurrence of REM sleep. (**d**) Frequency of brief awakenings (4–16 s) per hour of sleep for the first 2 hr after injections. (**e**) summary of effects of 5 and 10 mg/kg CNO on sleep in DREADD-free mice. n=10 for saline, n=6 for 1 mg/kg, n=10 for 5 mg/kg, n=8 for 10 mg/kg for spectral analysis. n=16 for saline, n=11 for 1 mg/kg, n=15 for 5 mg/kg, n=14 for 10 mg/kg for vigilance state analysis. n=15 for saline, n=11 for 1 mg/kg, n=15 for 5 mg/kg, n=13 for 10 mg/kg for analysis of brief awakenings. Asterisks in panels c and d indicate post hoc comparisons with significant differences (*p<0.05, **p<0.01, ***p<0.001,, ****p<0.001). Asterisks in panel a indicate frequency bins with significant differences in post hoc comparisons using uncorrected paired t-tests (p<0.05) following a significant interaction effect between 'frequency' and 'condition' in two-way ANOVAs. Data in panel a are presented as the mean ± s.e.m. (shaded areas). ANOVA: analysis of variance. CNO: clozapine-N-oxide. EEG: electroencephalogram. NREM: non-rapid eye movement sleep.

The online version of this article includes the following figure supplement(s) for figure 3:

**Figure supplement 1.** EEG power spectra of wakefulness, NREM, and REM sleep following injections of saline and 1 mg/kg CNO.

**Figure supplement 2.** EEG power spectra of wakefulness, NREM, and REM sleep following injections of saline and 5 mg/kg CNO.

**Figure supplement 3.** EEG power spectra of wakefulness, NREM, and REM sleep following injections of saline and 10 mg/kg CNO.

## C21 has sleep modulatory effects similar to CNO

Based on recent reports indicating that back-metabolism of CNO to clozapine causes behavioural effects in rodents (*Ilg et al., 2018*; *Gomez et al., 2017*; *Manvich et al., 2018*; *MacLaren et al., 2016*) and might affect sleep (*Varin et al., 2018*), we postulated that the effects of CNO injections on sleep could be avoided by using alternative DREADD ligands. However, another possibility is that it is the off-target binding of CNO to endogenous receptors that causes or contributes to the change in sleep patterns. In this scenario, the effects of CNO on sleep would be, at least in part, mediated by direct action of the DREADD actuator at neurotransmitter receptors, which are involved in the regulation of sleep and could not be overcome by minimising the conversion to clozapine. To discriminate between these two possibilities, we investigated whether the next-generation DREADD actuator C21 (*Chen et al., 2015*) has sleep modulatory effects similar to those observed after CNO injections. C21 does not back-convert to clozapine in vivo (*Thompson et al., 2018*) but has an almost identical profile of binding affinities to endogenous neurotransmitter receptors as CNO (*Jendryka et al., 2019*).

Intraperitoneal injections of C21 at a dose of 3 mg/kg modulated sleep compared to saline (*Figure 4* and *Figure 5*). The percentage of REM sleep was significantly reduced over the 6 hr observation period both in relation to the total sleep time ($t_{(6)}$ = 3.234, p=0.0089, Cohen's $d$=−1.2223, *Figure 4*) and to the recording time ($t_{(6)}$ = 2.086, p=0.0410, Cohen's $d$=−0.7885) with a strongly reduced REM to NREM ratio ($t_{(6)}$ = 3.253, p=0.0087, Cohen's $d$=−1.2296, *Figure 4—figure supplement 1*). The numerical reduction of REM sleep in the acute, 2 hr, time window following C21 injections did not reach statistical significance in this small sample of seven mice ($t_{(6)}$ = 1.829, p=0.0585, Cohen's $d$=−0.6972; *Figure 4b*), but had an effect size comparable to that of the statistically significant REM sleep reduction in the same time window for the 5 mg/kg CNO condition (5 mg/kg CNO vs. saline: Cohen's $d$=−0.5594).

In addition to the suppression of REM sleep, C21 elicited significant changes in sleep architecture (*Figure 5a and b*). The maximum and average duration of NREM sleep episodes was increased (maximum duration: $t_{(6)}$ = 2.551, p=0.0217, Cohen's $d$=0.9641; mean duration: $t_{(6)}$ = 2.462, p=0.0245, Cohen's $d$=0.9307) while the number of NREM sleep episodes in the first 6 hr following C21 injection was reduced ($t_{(6)}$ = 2.809, p=0.0154, Cohen's $d$=−1.0617; *Figure 5a*, *Supplementary file 5*). The same pattern was found in the acute observation time window (*Supplementary file 6*). The latency to NREM sleep was not significantly altered by C21 injections. For REM episodes there was no C21 effect on the maximum duration, however the mean REM episode duration was increased ($t_{(6)}$ = 2.082, p=0.0413, Cohen's $d$=0.7869), the number of REM episodes reduced ($t_{(6)}$ = 4.942, p=0.0013, Cohen's $d$=−1.8679) and the onset of REM sleep was delayed ($t_{(6)}$ = 2.043, p=0.0435, Cohen's $d$=0.7722, *Figure 5b*). As it was the case following CNO injections, the initial suppression of REM sleep following C21 injections prevented a meaningful analysis of REM architecture in the acute observation time window. For all NREM and REM architecture parameters that were significantly changed after CNO injections, effect sizes of 3 mg/kg C21 were similar to or slightly larger than those of 5 mg/kg CNO compared to saline (*Supplementary file 5*). EEG spectral analysis of NREM sleep indicated a temporary increase in low frequencies and a more pronounced and longer-lasting reduction in higher frequencies (*Figure 5c* and *Figure 5—figure supplement 1*) as observed in the 5 and 10 mg/kg CNO conditions (*Figure 3a*). The sample size for spectral analysis of the wake and REM spectra in the C21 condition was too low for statistical analysis (n=3) due to movement artefacts during wakefulness and sparseness of REM sleep. However, the qualitative changes of the EEG spectra analysed across all three vigilance states for this small sample suggest that the systematic spectral changes were specific to the NREM sleep spectrogram (*Figure 5—figure supplement 2*) as observed after 5 and 10 mg/kg CNO injections. State transition analysis indicated an increased probability to remain in the REM and NREM state (*Figure 5d*) and markers of sleep consolidation were also significantly altered as a result of C21 injections. The amount of NREM sleep before the first occurrence of REM sleep was increased ($t_{(6)}$ = 4.092, p=0.0032, Cohen's $d$=1.5467) and brief awakenings were reduced ($t_{(5)}$ = 2.164, p=0.0414, Cohen's $d$=−0.8836; *Figure 5e and f*).

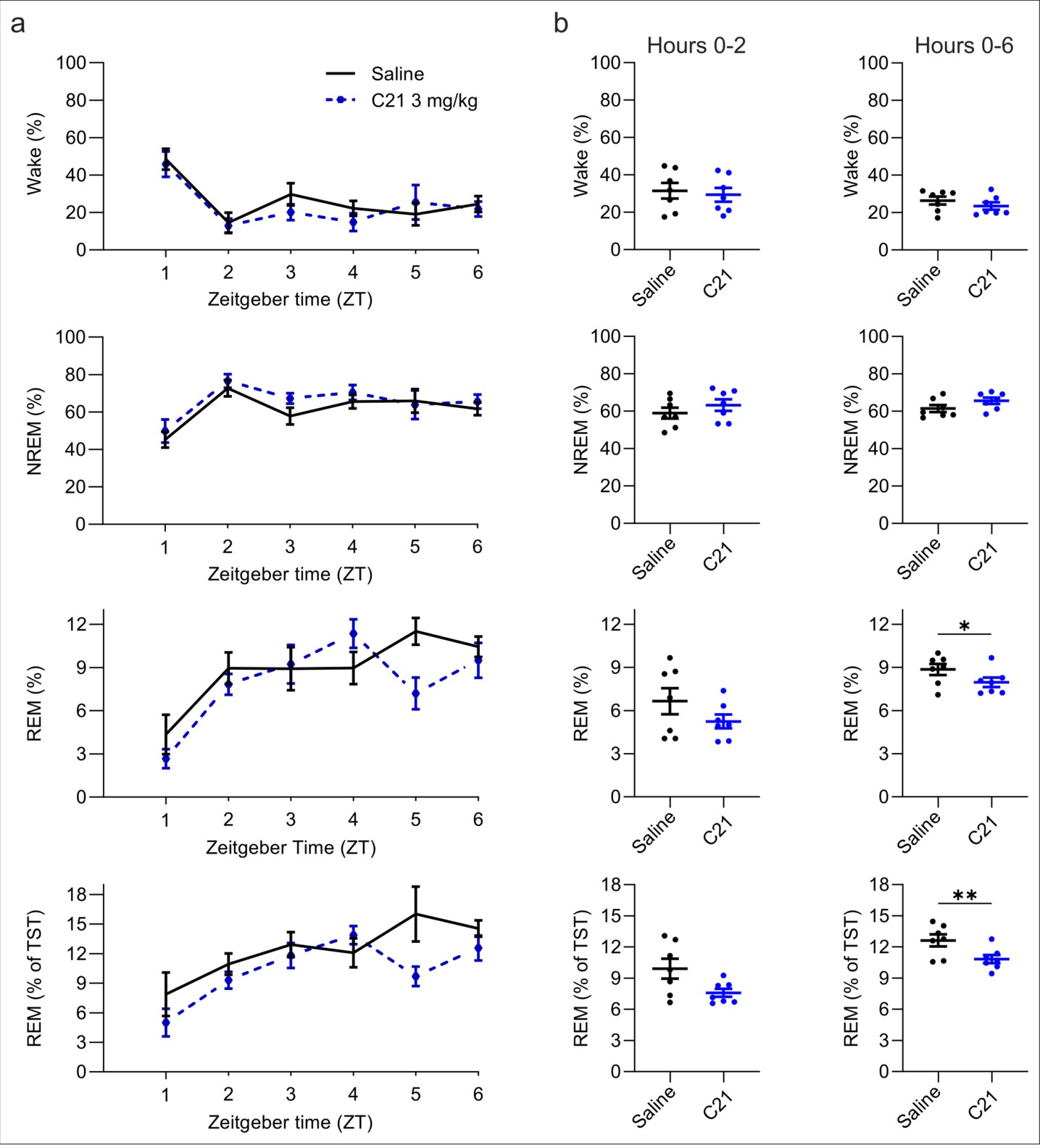

**Figure 4.** Suppression of REM sleep following C21 injections. (**a**) Time course of wakefulness, NREM, and REM sleep in the 6 hr following injection of C21 or saline at light onset (ZT 0). (**b**) Percentage of time spent in the three vigilance states during the first 2 hr (left column) and over the entire 6 hr observation period after saline and C21 injections. Note that REM sleep is presented both as proportion of the recording time (third row) and of the total sleep time (fourth row). n=7. Asterisks indicate t-tests with significant differences (*p<0.05, **p<0.01, ***p<0.001). C21: compound 21. NREM: non-rapid eye movement sleep. REM: rapid eye movement sleep. TST: total sleep time. ZT: zeitgeber time.

*Figure 4 continued on next page*

*Figure 4 continued*

The online version of this article includes the following figure supplement(s) for figure 4:

**Figure supplement 1.** Reduced REM/NREM ratio following C21 injections.

## Discussion

### CNO and C21 injections have clozapine-like effects on sleep

Our study demonstrates that the chemogenetic actuators CNO and C21 can modulate sleep in wild-type laboratory mice which do not express DREADD receptors. Both substances led to a suppression of REM sleep relative to NREM sleep, affected sleep architecture, and increased sleep state stability and sleep continuity consistent with more consolidated sleep (*Figure 3e*). The sleep changes following CNO and C21 injections in wild-type mice bore striking similarities with those previously reported for clozapine in rats (*Sorge et al., 2004*; *Spierings et al., 1977*) and humans (*Hinze-Selch et al., 1997*). In particular, the initial suppression of REM sleep and the occurrence of longer but fewer NREM episodes were the most consistent dose-dependent effects of clozapine on sleep in male Wistar rats, leading the authors to conclude that clozapine has sedative effects, suppresses REM initiation and increases sleep maintenance in rats (*Sorge et al., 2004*). Interestingly, in rats low doses of clozapine (2.5 mg/kg) had immediate sleep-promoting effects while high doses of clozapine (7.5 mg/kg) initially promoted wakefulness before a sustained increase in NREM sleep after the first 2 hr (*Sorge et al., 2004*). This might explain why in our study NREM latency was most strongly affected by the lowest dose (1 mg/kg) of CNO. Our findings of sleep modulatory effects of CNO are in line with previous reports of behavioural side effects of CNO resulting from back-metabolism to clozapine (*Gomez et al., 2017*; *Manvich et al., 2018*). However, our observation of similar sleep changes after injections with C21, a DREADD actuator that does not convert to clozapine (*Thompson et al., 2018*), suggests that in vivo metabolism to clozapine conversion is not the sole mechanism through which DREADD actuators can elicit unwanted effects.

### Possible mechanisms underlying sleep modulatory effects of CNO and C21

In our view, the most parsimonious explanation for the observed sleep modulatory effects of CNO and C21 is off-target binding at endogenous neurotransmitter receptors. It has been shown that CNO is a competitive inhibitor of several neurotransmitter receptors, including histaminergic H1, serotoninergic 5-HT$_{1A}$, 5-HT$_{1B}$, 5-HT$_{2A}$, 5-HT$_{2B}$, muscarinic M1, M2, M3, M4, adrenergic $\alpha_{1A}$ and $\alpha_{2A}$, and dopaminergic D1 and D2 receptors (*Gomez et al., 2017*; *Jendryka et al., 2019*). C21 has an off-target binding profile similar to CNO (*Jendryka et al., 2019*). In addition, a recent study reported increased firing rates of nigral dopaminergic neurons in wild-type rats, indicating that C21 can elicit direct neuromodulatory effects in rodents (*Goutaudier et al., 2020*). Among several endogenous receptors relevant for sleep regulation, CNO and C21 induce a strong competitive inhibition at histamine H1 receptors (*Jendryka et al., 2019*). Tested against a panel of G protein-coupled receptors, C21 had a greater affinity for histamine H1 receptors than for muscarinic DREADDs (*Thompson et al., 2018*). H1 receptor knockout and pharmacological antagonism of the H1 receptor in mice both result in a reduced number of brief awakenings, fewer but longer NREM sleep episodes, and a reduced latency to NREM sleep (*Huang et al., 2006*). In addition, antihistamines, which induce drowsiness, are known to strongly suppress REM sleep, but can be acutely NREM-promoting at low doses and wake-promoting at high doses, respectively (*Ikeda-Sagara et al., 2012*). Therefore, we speculate that the shared sleep modulatory effects of CNO and C21 might be in large part due to direct anti-histaminergic action. However, a contribution of other shared off-target sites of CNO and C21 such as the 5-HT$_{2A}$ receptors, which are thought to mediate the locomotor suppression after high doses of clozapine (*McOmish et al., 2012*), as well as anticholinergic effects leading to REM sleep suppression (*Jasper and Tessier, 1971*; *Niwa et al., 2018*), should also be taken into consideration.

### Previous indications for sleep modulatory effects of CNO

Many chemogenetic sleep studies provide adequate control data which exclude relevant effects of CNO on sleep in the given experimental paradigms (*Erickson et al., 2019*; *Mondino et al., 2021*; *Takata et al., 2018*; *Yu et al., 2019*; *Anaclet et al., 2015*; *Venner et al., 2016*). Many other studies do

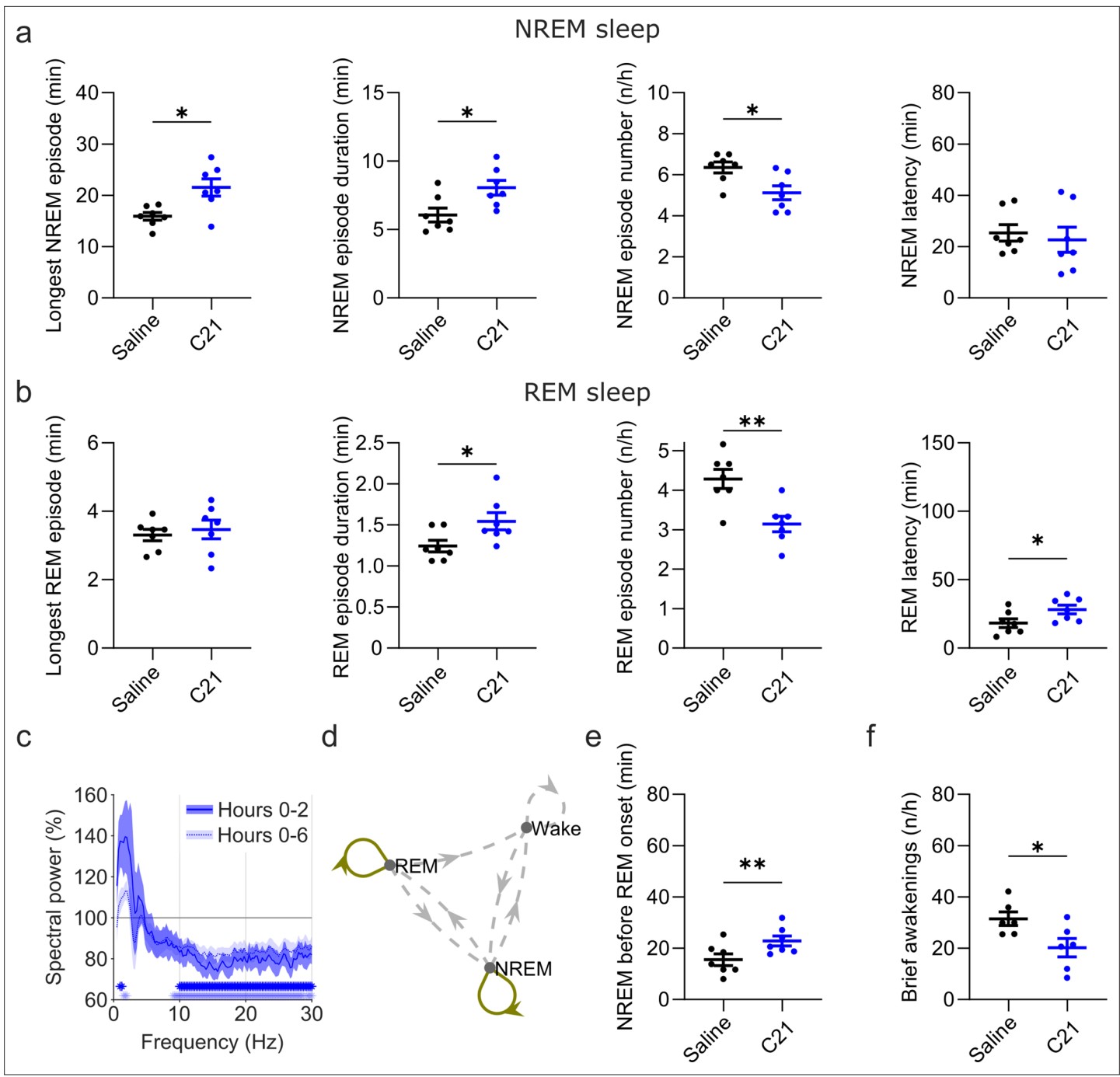

**Figure 5.** Effects of C21 on sleep architecture, NREM sleep spectra, sleep state stability, and sleep continuity resemble the effects of CNO. (**a**) NREM sleep architecture and (**b**) REM sleep architecture for the 6 hr observation period following C21 injections. (**c**) Frontal EEG spectra during NREM sleep relative to saline injections for the acute (first 2 hr, dark blue) and full (6 hr, light blue) observation period following C21 injections. Asterisks indicate frequency bins with significant differences in post hoc comparisons using uncorrected paired t-tests (p<0.05; acute: dark blue, full: light blue) following a significant interaction effect between 'frequency' and 'condition' in two-way ANOVAs. (**d**) Transitions between vigilance states in the 6 hr period following saline and C21 injections. Note the increased stability of REM and NREM sleep (REM>REM: p=0.0144, Cohen's d=1.0325; NREM>NREM: p=0.0384, Cohen's d=0.81527). Solid olive lines indicate significantly increased transitions/continuations of vigilance states in the C21 condition compared to the saline condition, dashed grey lines indicate all possible vigilance state transitions/continuations. (**e**) Cumulative amount of NREM sleep before the first occurrence of REM sleep. (**f**) Frequency of brief awakenings (4–16 s) per hour of sleep for the first 2 hr after injections. Number of animals n=7 mice for vigilance state analysis in panels a, b, d, and e. For analysis of EEG NREM spectra in panel c and brief awakenings in panel f: n=6 mice. Asterisks in panels a, b, e, and f indicate t-tests with significant differences (*p<0.05, **p<0.01, ***p<0.001). Data in c are presented as the mean ± s.e.m.

*Figure 5 continued on next page*

*Figure 5 continued*

(shaded areas). ANOVA: analysis of variance. C21: compound 21. EEG: electroencephalogram. NREM: non-rapid eye movement sleep. REM: rapid eye movement sleep.

The online version of this article includes the following figure supplement(s) for figure 5:

**Figure supplement 1.** EEG spectral analysis of NREM sleep following injections of saline and 3 mg/kg C21.

**Figure supplement 2.** EEG power spectra of wakefulness, NREM, and REM sleep following injections of saline and 3 mg/kg C21.

not present or analyse data of CNO-injected controls but instead only compare CNO vs. saline conditions in DREADD-expressing animals. However, occasionally authors have mentioned that the use of CNO injections of up to 10 mg/kg may have affected the sleep of animals in DREADD-free control groups (*Funk et al., 2017*) and recent work focussing on REM sleep regulation presented statistically significant effects of CNO doses of 5 mg/kg and above in the supplementary data (*Varin et al., 2018*). Another well-controlled study found a slight increase in NREM sleep bout duration of control mice injected with 0.3 mg/kg CNO, while all other analysed parameters were unaffected by this low CNO dose (*Liu et al., 2021*). This finding supports our effect size analysis indicating that NREM sleep bout duration is the sleep architectural parameter most strongly affected by CNO and C21 and that CNO can cause sleep-modulating effects at doses of 1 mg/kg CNO, and possibly below. Most studies that provide control data for injections of DREADD actuators in DREADD-free animals have small sample sizes, do not assess sleep architecture, absolute EEG spectra, or sleep continuity markers. In addition, differences in the zeitgeber time of drug application need to be considered when comparing our results to other studies. While sleep-modulating effects of drugs are usually assessed by injecting at light onset and assessing sleep during the early light period (*Kopp et al., 2002*; *Sorge et al., 2004*; *McKillop et al., 2021*; *Thomas et al., 2022*), the time of day when mice sleep most, sleep-modulating effects of DREADD actuators might be less pronounced when drugs are injected at dark onset and sleep is assessed during the early dark period when mice are typically active and sleep little (*Ferrari et al., 2022*). We speculate that the sleep-modulating effects of CNO and C21 have been overlooked in the past, because the change in the overall amount of sleep, and in the proportion of time spent in the respective vigilance states, is modest, and many studies only analysed these measures in their DREADD-free control groups. However, the effects of both DREADD actuators on the relative amount of REM sleep, sleep architecture, EEG power spectra of NREM sleep, sleep stability, and continuity are strong, robust, and highly relevant for future chemogenetic studies.

## Implications for future use of chemogenetics

Previous work has highlighted that several factors such as age, sex, and strain of the experimental animals (*Manvich et al., 2018*), as well as differences in the activity of the cytochrome P450 enzymes converting CNO to clozapine (*Mahler and Aston-Jones, 2018*), might contribute to the variability of side effects of DREADD actuators. In addition, the galenic formulation and use of solvents can alter the pharmacokinetic properties of DREADD actuators (*Campbell and Marchant, 2018*). The hydrochloride salt preparations of CNO used here and in other recent sleep studies (*Fernandez et al., 2018*; *Stucynski et al., 2021*) have reduced back-metabolism to clozapine and an improved water solubility and bioavailability compared to equivalent doses of CNO-DMSO preparations (*Allen et al., 2019*). The increased bioavailability and thereby supposedly amplified off-target receptor binding might explain that we find sleep-modulating effects of CNO already at doses where other studies have shown no effects. Considering the difficulty to predict behavioural effects of DREADD actuators in a specific experimental paradigm, our study supports the proposal to use the lowest dose of a DREADD actuator sufficient to elicit a DREADD-mediated effect in a respective experiment and to include a non-DREADD-expressing control group injected with the same dose of the respective actuator as the DREADD-expressing group in each individual experiment (*MacLaren et al., 2016*; *Mahler and Aston-Jones, 2018*; *Campbell and Marchant, 2018*). Most importantly, our work shows that avoiding clozapine back-metabolism of CNO by using novel DREADD actuators such as C21 does not prevent behavioural side effects. While it has been described that C21 can affect neuronal firing (*Goutaudier et al., 2020*), to our knowledge this is the first study demonstrating behavioural effects of a DREADD actuator that cannot convert to clozapine. In line with our finding, another next-generation DREADD agonist, perlapine, which is structurally similar to C21 (*Chen et al., 2015*;

*Thompson et al., 2018*), is long known as a REM-suppressing sedative and muscle relaxant used in the treatment of insomnia (*Ando et al., 1970*; *Allen and Oswald, 1973*; *Stille et al., 1973*). Due to the structural similarity between C21, perlapine and other novel chemogenetic actuators, such as deschloroclozapine (*Nagai et al., 2020*) and JHU37152 and JHU37160 (*Bonaventura et al., 2019*), systematically testing the impact of these actuators on sleep appears to be of paramount importance to validate their inertness in vivo. Chemogenetic approaches that do not require actuators with high chemical similarity to clozapine might provide an alternative for sleep research (*Magnus et al., 2019*), yet also their inertness in respect to neuronal activity and animal behaviour needs to be validated. Our work indicates that sleep analysis can reveal behavioural effects of DREADD agonists that are missed by other established behavioural tests such as the elevated plus maze and the marble burying task or measurements of locomotion and reaction time (*Jendryka et al., 2019*; *Tran et al., 2020*). Considering the complexity of neurotransmitter systems regulating sleep (*Saper and Fuller, 2017*) and the sensitivity of sleep architecture to pharmacological intervention (*Riemann and Nissen, 2012*), we propose that sleep assessment could serve as an invaluable tool to evaluate the biological inertness of newly developed chemogenetic actuators.

## Limitations

We would like to highlight that this study was initiated as a control experiment and was not designed to assess dose-dependent effects of CNO. Instead of counterbalancing all experimental conditions, we counterbalanced the saline and the 5 mg/kg CNO condition for the first two sessions to enable a direct comparison between saline and a medium dose of CNO avoiding potential adaptation effects resulting from repeated CNO injections. The decision to conduct a full study and to include C21 injections as an additional condition was only made once visual inspection of pilot data from four animals had indicated relevant effects of CNO on sleep. Because injections of 1 and 10 mg/kg CNO and 3 mg/ kg C21 were always performed after the initial two injections, sequence effects for those conditions cannot be excluded. This should particularly be considered in the interpretation of the results on NREM and REM sleep latency, which do not show a dose dependency. In order to reduce the number of animals used in laboratory research, most of the animals (13 out of 16) were used for combined experiments including other procedures such as light presentation with light-emitting diodes (LEDs) or local intracortical microinfusions. The type of experiments, the duration of the rest interval, and the within-subject design of our study make it unlikely that our findings were confounded by previous experiences but this cannot be fully excluded. Our power calculations indicate that even for a one-sided t-test a minimum sample size of n=11 would be required to detect large effects of $d$=1 with a power of 0.9 at the given α-error probability of 0.05. Although it is common to use sample sizes of 4–8 animals for DREADD-free control groups (*Hayashi et al., 2015*; *Funk et al., 2017*), sometimes comparing several drug doses (*Ferrari et al., 2022*), we consider the sample size of our C21 condition (n=7) as very small and suggest that our analyses concerning C21 effects should be considered exploratory. To avoid misinterpretation of one-sided null-hypothesis testing and to facilitate the comparison of our results with other studies, we provide effect sizes for all pairwise comparisons (*Supplementary file 5*) as well as an online repository with the raw data and analysis pipeline. It should also be highlighted that we only included a single dose of C21 (3 mg/kg) and therefore a dose-response assessment should be performed next to assess a putative dose dependency of C21 effects on sleep.

## Conclusions

In conclusion, our study suggests that the DREADD actuators, CNO and C21, have sleep modulatory effects, which cannot be explained by back-metabolism to clozapine alone but might result from off-target binding to endogenous receptors. This is the first demonstration that DREADD actuators that do not convert to clozapine can elicit relevant behavioural effects in DREADD-free animals. While our results require replication in an optimised experimental design, our findings have important implications for the future application of chemogenetics in sleep research and neuroscience. Our study highlights the need to use non-DREADD-expressing controls, even when novel actuators are used that cannot convert to clozapine. Considering the sensitivity of sleep architectural parameters to CNO and C21 demonstrated here, our work reveals a new opportunity of using simple EEG/EMG sleep screening to assess the pharmacological inertness of novel chemogenetic actuators in vivo. We are confident that these experimental refinements of the DREADD approach, plus novel technological

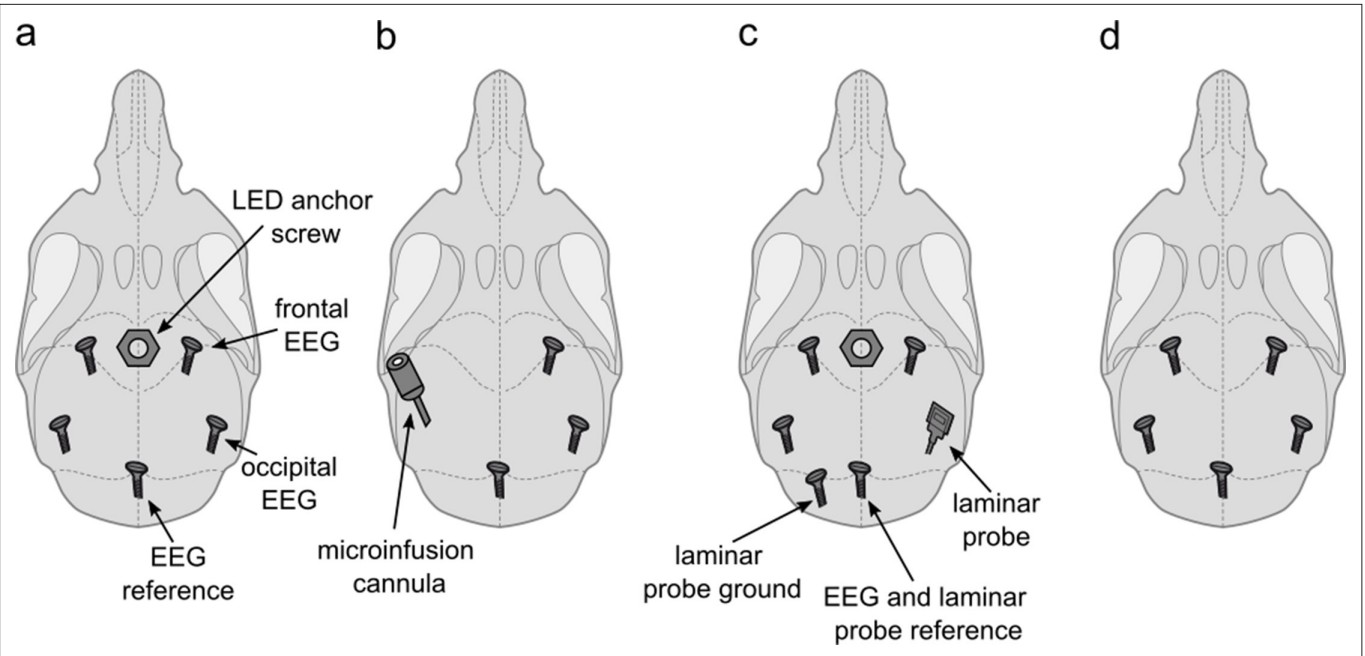

**Figure 6.** Implant configurations. (**a**) LED anchor screw, bilateral frontal and occipital EEG screws, and cerebellar reference screw. Implant configuration of n=5 mice. (**b**) Microinfusion cannula, right frontal and occipital EEG screws, and a cerebellar reference screw. Implant configuration of n=4 mice. (**c**) LED anchor screw, bilateral frontal and left occipital EEG screws, right occipital laminar probe, and cerebellar ground and reference screws. Implant configuration of n=4 mice. (**d**) Bilateral frontal and occipital EEG screws, and cerebellar reference screw. Implant configuration of n=3 mice. LED: light-emitting diode. EEG: electroencephalogram.

improvements, will unleash the full potential of this powerful tool in behavioural neuroscience and help pave the way for its clinical application.

## Materials and methods
### Animals
Sixteen young adult male C57BL/6J mice (age: 113±6 days, weight: 25.2±0.5 g) were used in this study. All animals were sourced internally from the Biomedical Services at the University of Oxford. As this study was originally designed as a control experiment, we did not intend to implant animals exclusively for this project. In order to reduce the number of animals used in laboratory research, 13 animals were implanted for combined sleep experiments involving procedures for this project and related work. All 16 animals used in this study were implanted with a right frontal EEG screw, a reference EEG screw above the cerebellum, and EMG wires in the neck muscles as described previously (***Fisher et al., 2016***). In addition to this EEG/EMG configuration, which provided the electrophysiological signals analysed in this manuscript, five of the animals were implanted with a frontal left and bilateral occipital EEG screws as well as with an anchor screw in the midline anterior to the frontal EEG screws, which served as a socket for a detachable LED device (***Figure 6a***); four animals were implanted with a right occipital EEG screw and a cannula (C315I; PlasticsOne) targeted to layer 5 of the primary somatosensory cortex (***Figure 6b***); four animals received an additional frontal and occipital EEG screw over the left hemisphere as well as a left cerebellar ground screw and a right occipital 16-channel laminar probe as well as a midline frontal anchor screw for the detachable LED device (***Figure 6c***); three animals received a left frontal and occipital EEG screw (***Figure 6d***). All animals had a rest interval of at least 3 days between the previous experiment and this study.

The nine animals implanted with a socket for the placement of a detachable LED device received flickering light stimulation to one of the eyes combined with a 4 hr sleep deprivation on up to 4 experimental days before being used for our study. All animals had at least 3 rest days without experimental interventions before the first injections of CNO or saline for the study presented here. The four animals implanted with a cannula in the left primary motor cortex received intracortical microinfusions

of bumetanide for a transient and localised blockade of the Na-K-2Cl cotransporter NKCC1 (*Kahle and Staley, 2008*), as well as of VU0436271 for a transient and localised blockade of the chloride potassium symporter KCC2 (*Sivakumaran et al., 2015*). Two of these animals underwent two intra-cortical microinfusions and the other two animals underwent three intracortical microinfusions before inclusion in this study. The last infusion took place 12 days before the start of the study presented here. Due to the limit of five injection or infusion procedures on the UK Home Office project license under which our experiments were conducted, these animals could only be subjected to three and two i.p. injections of different CNO doses and saline. The three animals implanted with bilateral frontal and occipital EEG screws were not used for any additional experiments. Only seven animals, the four animals implanted with laminar probes and the three animals used exclusively for this study, were used for C21 injections as this condition was added to the experimental protocol after the pilot data from CNO injections had been obtained.

## Electrophysiological signal acquisition, data processing, and sleep scoring

EEG/EMG recordings were performed using the 128 Channel Neurophysiology Recording System (Tucker-Davis Technologies Inc, Alachua, FL, USA) and the electrophysiological recording software Synapse (Tucker-Davis Technologies Inc, Alachua, FL, USA). Raw data was stored on a local computer in 24 hr recording blocks. During the continuous recordings EEG and EMG signals were filtered between 0.1 and 100 Hz, and stored at a sampling rate of 305 Hz. After transfer to an analysis desktop computer, the raw signals were resampled at a sampling rate of 256 Hz using custom-made code in MATLAB (The MathWorks Inc, Natick, MA, USA, version v2020a) and converted into the European Data Format (EDF) as previously described (*McKillop et al., 2018*). The EDF files were visually scored in individual 4 s epochs by blinded experimenters using the software package Sleep Sign for Animals (SleepSign Kissei Comtec Co., Ltd., Nagano, Japan). If EEG signals contained temporary artefacts due to electrical noise, movements, or chewing, the respective vigilance state was assigned to the respective epoch but the EEG signals were not included in the spectral analysis. Using the Sleep Sign for Animals software, fast Fourier transform routine (Hanning window) with a 0.25 Hz resolution was computed in the frequency range between 0 and 30 Hz for each individual 4 s epoch.

## CNO and C21 products

To avoid the use of the toxic solvent dimethyl sulfoxide (DMSO), which is typically used for the preparation of CNO products in concentrations of up to 15% (*Campbell and Marchant, 2018*), we opted for the use of a water-soluble salt preparation of CNO, CNO dihydrochloride (Tocris, Bio-Techne LTD, Abingdon, UK, catalog no.: 6329) dissolved in sterile saline. The dihydrochloride preparation of CNO undergoes less back-metabolism to clozapine but has a higher bioavailability compared to CNO-DMSO as indicated by pharmacokinetic work in rhesus macaques (*Allen et al., 2019*). This product has previously been used in sleep studies on mice at concentrations between 1 and 5 mg/kg (*Fernandez et al., 2018*; *Stucynski et al., 2021*). For C21 injections we used the water-soluble version of DREADD agonist C21 (C21 dihydrochloride, Tocris, Bio-Techne LTD, Abingdon, UK, catalog no.: HB6124). We chose a dose of 3 mg/kg because a detailed pharmacokinetic assessment of this product at this specific concentration as well as behavioural testing in a five-choice serial-reaction-time task did not reveal any behavioural effects at this dose (*Jendryka et al., 2019*).

## Experimental design

All 16 animals received a medium dose of CNO (5 mg/kg, 0.25 mg/ml solution) and an equivalent volume of saline in a counterbalanced order with a 72 hr rest interval between the first two experimental sessions. Of those 16 mice, 14 also received a 10 mg/kg CNO injection (0.5 mg/ml solution) and 12 a 1 mg/kg CNO injection (0.05 mg/ml solution). These two injections were counterbalanced as third and fourth injections. Seven animals also received a 3 mg/kg (0.15 mg/ml) C21 injection in a fifth experimental session. This semi-counterbalanced design was chosen to ensure that at the medium dose condition is counterbalanced with saline injections to circumvent putative habituation or adaptation effects following repeated injection of CNO.

Animals were on a regular light-dark cycle (lights on at 9 am) and intraperitoneal (i.p.) injections were performed within 15 min after light onset following a brief health check. The delay between

light onset and injection was (mean ± s.e.m.): saline: 4.19±0.87 min, 1 mg/kg CNO: 4.58±0.59, 5 mg/kg CNO: 3.52±1.04 min, 10 mg/kg CNO: 4.75±0.59 min, 3 mg/kg C21: 6.01±0.96 min. Individual recording sessions were separated by a rest interval of at least 72 hr. The recording chambers were kept open for approximately 10–15 min after the injection to monitor for potential adverse effects. The chambers were then closed and the animals checked remotely at regular intervals for the first 6–12 hr after injection. For data analysis, all recordings were aligned to the time point of injections.

## Sample size determination and power analysis

Sample size and power calculations were performed using G*Power 3.1, an open-source statistical power analysis program (*Faul et al., 2007*). The sample size was chosen based on previous experiments in our lab investigating the effects of the sedative diazepam on sleep (*McKillop et al., 2021*), which indicated an effect size of Cohen's $d$=0.90 for the key outcome parameter NREM sleep time. We therefore decided that our study should be sufficiently powered to detect effects of sizes of $d$=1 designed between saline and individual CNO treatment conditions with a power of 0.9 at the given α-error probability of 0.05. The estimated sample size from this calculation was 12–13 animals per group. Based on experiences from previous EEG studies in mice, we aimed to account for an attrition rate of approximately 20% and decided to include 16 animals in this study.

## Statistical procedures

Data were analysed using MATLAB (version R2020a; The MathWorks Inc, Natick, MA, USA), SAS JMP (version 7.0; SAS Institute Inc Cary, NC, USA), and IBM SPSS Statistics for Windows (version 25.0; IBM Corp., Armonk, NY, USA). Reported averages are mean ± s.e.m. For all analyses a significance level of p=0.05 was adopted. For the statistical comparison of CNO and saline injections, mixed-effect models were calculated using GraphPad Prism (version 9.1.1 for Windows; GraphPad Software, San Diego, CA, USA, https://www.graphpad.com/). Significant main effects of treatment conditions were followed up with Dunnett's adjustment for post hoc comparisons. For time courses, we applied mixed-effect models to the acute (0–2 hr post injection) and prolonged (0–6 hr post injection) time window separately. For spectral analysis, EEG power spectra of individual animals were log-transformed before hypothesis testing. We performed two-way ANOVAs (analyses of variance) with the factors 'frequency' and 'condition' and conducted post hoc tests for individual frequency bins only when a significant interaction effect between 'frequency' and 'condition' was found. For the post hoc tests, individual spectral bins were compared between individual CNO/C21 treatments and saline in three different ways to guarantee the most informative and unbiased illustration of relevant frequency ranges. We first used uncorrected two-tailed t-tests for paired samples at an α-error threshold of p=0.05. No correction was applied in these cases because the 119 EEG spectral bins do not vary independently and hence corrections for multiple comparisons can be considered too conservative and reduce statistical power (*Achermann and Borbély, 1998*). However, to assess the robustness of differences in the EEG power spectra between CNO/C21 conditions and saline, we also performed post hoc tests using uncorrected two-tailed t-tests for paired samples at an α-error threshold of p=0.01 and validated our results performing Benjamini-Hochberg correction for multiple testing (*Jafari and Ansari-Pour, 2019*; *Benjamini and Hochberg, 1995*). Comparisons between the C21 and saline condition were performed using one-tailed t-tests to assess whether sleep variables were changed in the same direction as after CNO treatment. For analysis of the percentage of time spent in the three vigilance states, wakefulness, NREM sleep and REM sleep were expressed as the percentage of time in the respective time window, REM was further expressed as percentage of total sleep time in the respective time window as in previous work (*Huber et al., 1999*; *Kashiwagi et al., 2020*). The REM/NREM ratio is also presented in supplementary figures. For sleep architecture analysis, wake and NREM sleep episodes were defined as intervals of at least 1 min allowing an interruption of 4–16 s as in previous work (*Krone et al., 2021*). Considering that REM episodes are on average considerably shorter than wake or NREM episodes in mice (*McShane et al., 2010*; *Huber et al., 2000*), REM sleep episodes were defined as intervals of at least 16 s allowing an interruption of 4–8 s. Brief awakenings were defined as up to 16 s interruptions of sleep by wake-like EEG and EMG patterns (*Franken et al., 1991*). We performed a state transition analysis to assess the shifts from and continuations of a vigilance state for the first 6 hr following injections. Shifts and continuations were defined as the relative percentage of shifts and continuations per vigilance state. Sleep stage-transition probabilities were analysed using

non-parametric paired tests performed on the stage-transition occurrences using bootstrap statistics (5000 iterations). A p-value was defined as the number of instances where the value obtained from random sampling was larger than that observed in the data divided by the number of iterations. In this way, we could calculate the probability of obtaining our results by chance and control the familywise error rate (*Maris and Oostenveld, 2007*; *Pernet et al., 2015*). In all figures, significance levels of post hoc comparisons are indicated with black asterisks: '*' for 0.05≥ p > 0.01; '**' for 0.01≥ p > 0.001; '***' for 0.001 ≥ p; '****' for 0.0001 ≥ p. Effect sizes are reported as Cohen's *d* calculated using the MATLAB function computeCohen_d (Ruggero G Bettinardi (2020). computeCohen_d(x1, x2, varargin) (https://www.mathworks.com/matlabcentral/fileexchange/62957-computecohen_d-x1-x2-varargin), MATLAB Central File Exchange. Retrieved 4 October 2020). Data from one animal had to be partially excluded because of a defective EEG headstage. This animal had to be excluded from the analysis of brief awakenings due to EMG artefacts, which made it difficult to identify a sudden increase of muscle tone during sleep. The same animal also had to be excluded from the analysis of the 1 and 5 mg/kg CNO treatments because of technical issues. For spectral analysis six animals were originally excluded due to occasional artefacts in the EEG signals. However, prompted by a reviewer comment during the revision process of this manuscript, we have included three of these animals in the spectral analysis of NREM sleep in the C21 condition after careful visual checks of individual EEG spectrograms for each animal and vigilance state because EEG artefacts were largely restricted to the wake state when the animals moved.

## Acknowledgements

We thank Prof. Dr Dennis Kätzel from the University of Ulm, Germany, for his advice on the study design and the preparation of the DREADD actuators; Prof. Dr Antoine Adamantidis from the University of Bern, Switzerland, for his comments on the manuscript; all members of the laboratory of the Vyazovskiy lab for kind help with surgery assistance and animal care. This work was supported by a travel scholarship from the German Academic Exchange Service (DAAD) and a studentship from the Studienstiftung des Deutschen Volkes awarded to JT as well as a Wellcome Trust PhD studentship 203971/Z/16/Z to LBK. LBK was also supported by a Mann Senior Scholarship in medical sciences by Hertford College, Oxford. LEM was supported by a Novo Nordisk Postdoctoral Fellowship run in partnership with the University of Oxford. LEM was also supported by a Sir Paul Nurse Junior Research Fellowship at Linacre College, Oxford. HA was supported by a Wellcome Trust Postdoctoral Fellowship (206500/Z/17/Z). The laboratory of ZM received funding from the UK Medical Research Council (G00900901), the Royal Society, St John's College Research Centre, the Anatomical Society and Einstein Stiftung. ZM is an Einstein Visiting Fellow at Charité-Universitätsmedizin Berlin (host B Eickholt for 2020–2024), and lead researcher at Oxford Martin School, University of Oxford. This work was further supported by a Wellcome Trust Strategic Award (098461/Z/12/Z), a John Fell OUP Research Fund grant (131/032) and Medical Research Council (UK) grants MR/N026039/1 and MR/S01134X/1.

## Additional information

### Funding

| Funder | Grant reference number | Author |
| --- | --- | --- |
| Wellcome Trust | 203971/Z/16/Z | Lukas B Krone |
| Medical Research Council | G00900901 | Zoltan Molnar |
| Wellcome Trust | 098461/Z/12/Z | Vladyslav V Vyazovskiy |
| Oxford University Press | 131/032 | Vladyslav V Vyazovskiy |
| Medical Research Council | MR/N026039/1 | Zoltan Molnar |
| Medical Research Council | MR/ S01134X/1 | Vladyslav V Vyazovskiy |

| Funder | Grant reference number | Author |
|---|---|---|
| Studienstiftung des Deutschen Volkes | | Janine Traut |
| Hertford College, University of Oxford | | Lukas B Krone |
| Linacre College, University of Oxford | | Laura E McKillop |
| Novo Nordisk UK Research Foundation | | Laura E McKillop |
| Deutscher Akademischer Austauschdienst | | Janine Traut |
| St. John's College, University of Oxford | | Zoltan Molnar |
| Einstein Stiftung Berlin | | Zoltan Molnar |
| Anatomical Society | | Zoltan Molnar |
| Wellcome Trust | 206500/Z/17/Z | Hannah Alfonsa |

The funders had no role in study design, data collection and interpretation, or the decision to submit the work for publication. For the purpose of Open Access, the authors have applied a CC BY public copyright license to any Author Accepted Manuscript version arising from this submission.

## Author contributions

Janine Traut, Conceptualization, Data curation, Formal analysis, Validation, Investigation, Visualization, Methodology, Writing – original draft, Writing – review and editing; Jose Prius Mengual, Data curation, Formal analysis, Validation, Investigation, Methodology, Project administration; Elise J Meijer, Data curation, Validation, Investigation, Methodology; Laura E McKillop, Resources, Data curation, Formal analysis, Supervision, Validation, Investigation, Methodology; Hannah Alfonsa, Resources, Data curation, Validation, Investigation, Methodology; Anna Hoerder-Suabedissen, Resources, Supervision, Validation, Investigation, Visualization, Methodology, Project administration; Seo Ho Song, Software, Formal analysis, Validation, Methodology; Kristoffer D Fehér, Software, Formal analysis, Visualization, Methodology; Dieter Riemann, Formal analysis, Supervision, Funding acquisition, Validation, Methodology, Writing – review and editing; Zoltan Molnar, Conceptualization, Resources, Supervision, Funding acquisition, Validation, Writing – original draft, Project administration, Writing – review and editing; Colin J Akerman, Conceptualization, Resources, Supervision, Funding acquisition, Validation, Methodology, Writing – original draft, Project administration, Writing – review and editing; Vladyslav V Vyazovskiy, Conceptualization, Resources, Software, Formal analysis, Supervision, Funding acquisition, Validation, Methodology, Writing – original draft, Project administration, Writing – review and editing; Lukas B Krone, Conceptualization, Resources, Data curation, Software, Formal analysis, Supervision, Funding acquisition, Validation, Investigation, Visualization, Methodology, Writing – original draft, Project administration, Writing – review and editing

## Author ORCIDs

Janine Traut ![ORCID] http://orcid.org/0000-0002-2169-2670
Jose Prius Mengual ![ORCID] http://orcid.org/0000-0002-4089-5254
Elise J Meijer ![ORCID] http://orcid.org/0000-0002-4812-3275
Laura E McKillop ![ORCID] http://orcid.org/0000-0003-3085-1175
Hannah Alfonsa ![ORCID] http://orcid.org/0000-0002-6357-7494
Anna Hoerder-Suabedissen ![ORCID] http://orcid.org/0000-0003-1953-7871
Seo Ho Song ![ORCID] http://orcid.org/0000-0003-2970-2746
Kristoffer D Fehér ![ORCID] http://orcid.org/0000-0002-1205-7157
Dieter Riemann ![ORCID] http://orcid.org/0000-0002-1968-6220
Zoltan Molnar ![ORCID] http://orcid.org/0000-0002-6852-6004
Colin J Akerman ![ORCID] http://orcid.org/0000-0001-6844-4984
Vladyslav V Vyazovskiy ![ORCID] http://orcid.org/0000-0002-4336-6681
Lukas B Krone ![ORCID] http://orcid.org/0000-0002-5535-7221

## Ethics

All experiments were performed in accordance with the United Kingdom Animal Scientific Procedures Act 1986 under personal and project licences granted by the United Kingdom Home Office. Ethical approval was provided by the Ethical Review Panel at the University of Oxford. Animal holding and experimentation were located at the Biomedical Sciences Building (BSB) and the Behavioural Neuroscience Unit (BNU), University of Oxford. Data was collected and reported in accordance with the ARRIVE guidelines. All procedures were optimised to minimise the pain, suffering, distress or lasting harm that research animals might experience.

## Decision letter and Author response

Decision letter https://doi.org/10.7554/eLife.84740.sa1
Author response https://doi.org/10.7554/eLife.84740.sa2

---

# Additional files

## Supplementary files

• Supplementary file 1. Time spent in wakefulness, non-rapid eye movement (NREM), and rapid eye movement (REM) sleep after clozapine-N-oxide (CNO) and saline injections.

• Supplementary file 2. Sleep architecture after clozapine-N-oxide (CNO) and saline injections.

• Supplementary file 3. Non-rapid eye movement (NREM) sleep architecture for the first 2 hr after clozapine-N-oxide (CNO) and saline injections.

• Supplementary file 4. Analysis of variance (ANOVA) results for electroencephalography (EEG) spectra following clozapine-N-oxide (CNO) and saline injections.

• 1Supplementary file 5. Sleep parameters after saline and compound 21 (C21) injections.

• Supplementary file 6. Non-rapid eye movement (NREM) sleep architecture for the first 2 hr after compound 21 (C21) and saline injections.

• MDAR checklist

## Data availability

This article is accompanied by a figshare project containing detailed information about the experiment (https://doi.org/10.6084/m9.figshare.21507561), pre-processed raw data (DOI: https://doi.org/10.6084/m9.figshare.21507567), custom written MATLAB analysis scripts (https://doi.org/10.6084/m9.figshare.21507573) and functions (https://doi.org/10.6084/m9.figshare.21507576) as well as GraphPad Prism files (https://doi.org/10.6084/m9.figshare.21507570) with source data, analysis sheets, and individual figure panels.

The following datasets were generated:

| Author(s) | Year | Dataset title | Dataset URL | Database and Identifier |
|---|---|---|---|---|
| Krone L | 2023 | Summary table with information about the experiment | https://doi.org/10.6084/m9.figshare.21507561.v1 | figshare, 10.6084/m9.figshare.21507561.v1 |
| Krone L | 2023 | Individual EEG recordings | https://doi.org/10.6084/m9.figshare.21507567.v1 | figshare, 10.6084/m9.figshare.21507567.v1 |
| Krone L | 2023 | Matlab analysis scripts | https://doi.org/10.6084/m9.figshare.21507573.v1 | figshare, 10.6084/m9.figshare.21507573.v1 |
| Krone L | 2023 | Matlab Functions | https://doi.org/10.6084/m9.figshare.21507576.v1 | figshare, 10.6084/m9.figshare.21507576.v1 |
| Krone L | 2023 | GraphPad Prism files with source data, analysis sheets, and individual figure panels | https://doi.org/10.6084/m9.figshare.21507570.v1 | figshare, 10.6084/m9.figshare.21507570.v1 |

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
