## [Editor Report]

This study uses EMG/EEG recordings to demonstrate that the DREADD actuators CNO and C21 have sleep modulatory effects that might result from off-target binding to endogenous receptors. This study highlights the need to use non-DREADD-expressing controls, even when novel actuators are used that cannot convert to clozapine.

---

## [Decision Letter]

**Decision letter after peer review:**

[Editors’ note: the authors submitted for reconsideration following the decision after peer review. What follows is the decision letter after the first round of review.]

Thank you for submitting the paper "Effects of clozapine-N-oxide and compound 21 on sleep in laboratory mice" for consideration by *eLife*. Your article has been reviewed by 2 peer reviewers, and the evaluation has been overseen by a Reviewing Editor and a Senior Editor. The reviewers have opted to remain anonymous.

Comments to the Authors:

We are sorry to say that, after consultation with the reviewers, we have decided that this work, in its current form, will not be considered for publication by *eLife*. I am sorry that we could not be more positive on this occasion regarding publication in *eLife*, but we hope that you find our referees' comments helpful. As you can see, the two reports highlight shared concerns regarding the statistical robustness of the findings. It seems that a significant amount of additional work would be needed to address these concerns. Regrettably, the manuscript cannot be accepted without additional experiments that are needed to substantiate the conclusions. However, we will be happy to reconsider a new manuscript that fully addresses the reviewer's critique.

*Reviewer #1 (Recommendations for the authors):*

Traut, Mengual et al. investigated the impact of the chemogenetic ligands clozapine N-oxide (CNO) and compound 21 (C21) on sleep in wildtype C57Bl/6J mice. They report that intraperitoneal injection of three CNO doses (1, 5, and 10 mg/kg) dose-dependently suppresses rapid eye movement (REM) sleep, consolidates non-rapid eye movement (NREMS), and alters NREM electroencephalographic (EEG) spectra based on standard EEG/electromyographic (EMG) assessment of sleep-wake behavior. The sleep field has generally attributed CNO effects on sleep to back-metabolism of CNO to sleep-promoting clozapine; however, the authors also report similar sleep effects by C21 administration-a chemogenetic ligand that does not back-metabolize to clozapine. Therefore, the authors reason that the impact of chemogenetic ligands on sleep could also be due to off-target effects on endogenous neurotransmitter systems. This paper highlights the need for important non-DREADD expressing controls in sleep research, but this conclusion can be extended to other neuroscience research using chemogenetics. This conclusion is supported by the data using techniques that are easily implemented by those in the sleep field, but there are some aspects of analysis that can be improved.

1) The authors report post-hoc significance for statistical tests that did not have significant main effects (Figures 1b, 2c). In addition, no corrections for multiple testing of spectral data are implemented. The authors are transparent about this reporting, however.

2) Analyses of REM sleep can be improved. Time spent in REM sleep is reported as a percentage of total sleep time which means that REM sleep is being assessed using different comparators across vehicle and CNO doses. Additionally, REM sleep architecture is based on REM sleep episodes {greater than or equal to} 1 min which tends to be longer than an average REM sleep episode in a mouse.

3) Authors acknowledge that a limitation of this study is that many of the animals included were intended for different experimental purposes and underwent other procedures prior to this study. However, the within-subject design and rest periods prior to this study should reduce potential impacts of prior experimental procedures on the present findings.

Recommendations

1) Although I appreciate the authors' transparency, there are a few corrections that should be made in terms of statistical reporting:

a. Do not round the degrees of freedom reported for the F statistic in the text.

b. If there are no significant main effects, post-hoc analyses should not be initiated. Therefore, the post-hoc data reporting and significance markers should be removed for Figures 1b and 2c.

c. It is not clear if a repeated measures ANOVA (or a similar test) was run prior to pairwise comparisons of spectral data. Also, these pairwise comparisons need to be corrected for multiple testing using something like a Benjamini-Hochberg correction. One option to help with spectral analyses is to reduce the resolution to 0.5 or 1 Hz without compromising the integrity of the data.

d. The authors describe post-hoc contrasts with regard to t-tests for C21 analyses.

2) What is the rationale for reporting time spent in REM sleep as a percentage of total sleep time? This analysis uses different comparators (total NREM + REM vs total recording time) across conditions. I think it would be better to report REM sleep as a percentage of total recording time in Figures 1 and 4 as is reported for wakefulness and NREM sleep to get a more direct assessment of REM sleep suppression. Reporting REM sleep as a percentage of total sleep time can still be useful for assessing NREM/REM ratios.

3) Using {greater than or equal to}1 min to define a REM episode seems quite long for a mouse (see McShane et al., J Neurosci Methods, 2010). This criterion could be excluding a majority of REM episodes.

4) Many of the figures contain both red and green which makes it difficult for some readers to differentiate between conditions, especially in Figures 1a, 2a, and 3a. Please switch to a more inclusive color palette. In addition, it is difficult to see the 5 mg/kg data in Figure 3a despite that data being the primary focus of the text.

5) Page 8, Line 157 seems to state in error that "REM sleep episodes were significantly shorter…"

6) Because sleep impacts a vast number of biological processes and behaviors, it would be helpful to briefly include in the Discussion how these findings can impact the broader neuroscience community beyond the sleep field. Also, it might be helpful to include what is known (if anything) on chemogenetic ligand administration during the dark period.

*Reviewer #2 (Recommendations for the authors):*

The use of DREADDs in sleep research is common. In typical experiments, it is assumed that the DREADD actuators on their own do not affect sleep, and a typical control group may receive saline i.p. injections that are considered adequate to control for the effects of injection/animal handling on sleep. However, here the authors show that intraperitoneal application of the common DREADD actuators Clozapine-N-oxide (CNO) and compound 21 (C21) modulate sleep architecture and sleep EEG features compared to saline. Specifically, CNO reduced REM sleep, led to longer bouts of NREM sleep and elevated EEG power in the lower frequencies at the expense of power > 6Hz.

The experiments and analysis are carefully conducted and the manuscript is clearly written. The results are nicely put in the context of existing literature and their implications are well discussed. In addition, the use of C21 elegantly addresses the issue of clozapine back-metabolism. At the same time, it is evident that the authors started this investigation as a control experiment for another project. As such, the experimental design and data acquisition protocols are suboptimal to address the question at hand. For example, different experiments and analyses have a different number of animals, and conditions are not necessarily counterbalanced in their order. This diminishes statistical power (since comparisons are performed between a limited group of subjects, rather than within subjects). While this work is important for the research community, it is quite technical and the overall impression is that results are modest and may not be very robust.

While this work is important for the research community, my enthusiasm is dampened by two issues:

1. The overall impression is that the results are modest and not entirely robust. Different effects are observed for different drug doses and different features of sleep (e.g. NREM is longer but with fewer bouts, but no effect on overall % of NREM; some effects such as delayed transitions to REM sleep are strongest with low doses; EEG spectrogram effects are most evident at middle doses). Given the many tests employed (NREM/REM * different features such as number of bouts or their duration * many temporal intervals examined separately * 3 doses) there may be a multiple comparison challenge that, together with the limited number of animals in some experiments (n=3 EEG spectral analysis in C21), and some statistical issues described below, raise some doubts about whether robust conclusions can be drawn.

2. The other issue is that I find this report very technical. Although important for investigators in the field, it may not appeal to a wide audience.

With the combination of these two issues, I don't think that the manuscript in its present form is suitable for *eLife*.

Specific comments:

– To be best of my knowledge, it may not be appropriate to perform Post-hoc statistical tests if initial ANOVA is not significant. Many other groups/papers would consider these results non-significant and not report them.

– I wonder how the results would seem if authors were to use only the 11 mice that had all 4 conditions? I understand that power calculations indicate that 12-13 mice are necessary, but using a within-subjects approach could serve as a complementary approach with potential to yield more robust findings.

– C 21 data: sample seems too small (especially for the spectral analysis with only n=3 mice) and lack dose dependency analysis. This makes it difficult to compare effect sizes for CNO and C21.

– There are places where statistical approach may suffer from a multiple comparison issue. For example, in figure 1 the significance threshold is set at 0.05 but the % in each state (wake/NREM/REM) is evaluated for each dose/condition, and for multiple temporal intervals (six 1h bins). Consequently, the minimal p value may have to be adjusted.

– Line 157: "REM sleep episodes were significantly shorter following CNO injections" but in Figure 2C second panel from right it seems that REM episodes are longer.

– While most results are dose-dependent latency effects are strongest for lowest dose, which seems strange, even though the authors provide some thoughts in the discussion.

– Figure 2- the color green is both wake (in panel a) and 1 mg/kg (throughout the paper). To avoid confusion, it could be best to select different colors.

– In line 184 it is mentioned that the comparison was between 5 mg/kg and saline. You show all 3 doses in the main figure and in the supplementary you show all 3 states and not only NREM as described in text.

– In line 194 you point to figure 1, but perhaps it is mean to point to figure 3a?

– (methods) If I have it right either EEG or LFP were used for spectral analysis. This inconsistency likely adds to the variability.

– (methods) Comparisons between EEG ranges may require correction for multiple comparisons.

– (methods) Using 1 tailed t-test for the C21 experiments seems strange, although the logic is explicitly explained.

– (methods) It seems that not all excluded mice are accounted for.

[Editors’ note: further revisions were suggested prior to acceptance, as described below.]

Thank you for resubmitting your work entitled "Effects of clozapine-N-oxide and compound 21 on sleep in laboratory mice" for further consideration by *eLife*. Your revised article has been evaluated by John Huguenard (Senior Editor) and a Reviewing Editor.

The manuscript has been improved but there are some remaining issues that need to be addressed, as outlined below:

Essential revisions:

1) Please add ANOVA prior to pairwise testing – it may also prevent unnecessary pairwise testing that would reveal a few significant scattered spectral bins that do not reflect a systematic change that the authors detail in the remainder of their response which focuses on multiple testing corrections for pairwise tests and bin size considerations.

2) Related, the authors refer to the pairwise testing of EEG spectral data as "post-hoc" testing, but statistically speaking, none of these comparisons are "post-hoc" because ANOVAs were not conducted prior to pairwise testing. See Figure 3, Figure 5, Figure 3 —figure supplement 1, Figure 3 —figure supplement 2, Figure 3 —figure supplement 3, and Figure 5 —figure supplement 1.

Please consider additional points for discussion as suggested by the reviewers.

*Reviewer #1 (Recommendations for the authors):*

The authors did a comprehensive job in addressing the concerns raised in the first round of review. The manuscript improved significantly. I only have a few final suggestions, which are all minor; the authors should decide on whether and how to incorporate them into the final manuscript:

1. In their response, the authors write that "While we agree with the reviewer that the effect of CNO and C21 on the overall amount of sleep could be considered modest – and therefore sleep-modulating effects of these actuators have often been missed in the past – our results that … are strong and robust."

I think that the authors may consider conveying this message also to the readers e.g. in Discussion. The fact that some effects can be considered modest, yet consistent and of potential important physiological significance, may be important to explicitly acknowledge in terms of relation to previous studies and as something for future chemogenetic studies for all the community.

2. It could be worthwhile to separately show the effects during the first 2h in Figure 2. Most people/studies employing chemogenetics only look at first few hours and this may be most relevant for them.

3. It could be worthwhile that limitations section in Discussion includes that C21 only had one dose (3 mg/kg) rather than a more complete dose-response assessment (judging by the CNO data, effects may vary depending on dose so this may be important)

4. In Discussion lines 420-> the authors suggest that "it is vital to minimize the injected doses of DREADD actuators" but sometimes lower or medium doses have stronger effects, so this statement may be overly simplistic

*Reviewer #2 (Recommendations for the authors):*

This resubmission by Traut, Mengual et al. is much improved. The additions and edits now provide a more complete and accurate presentation of the data especially for REMS time and architecture.

Including REMS time as a % of total recording time (TRT) now provides a more informed understanding of REMS as % of total sleep time (TST) because it provides context for the TST metric (i.e., is NREMS changing, is REMS changing, or are both states changing). Reporting both metrics is how other groups the authors cited (Franken, Malafosse, and Tafti 1999; Huber, Deboer, and Tobler 1999; Kashiwagi et al. 2020) approached the analysis as well. We can now see that the impact of CNO on REMS is nuanced preserving overall REMS amounts but reducing REMS relative to NREMS for higher doses of CNO. It might be helpful to include the REMS as a % of TRT data as the third row in Figures 1 and 4 (after NREMS and before REMS as a % of TST) as the papers cited above did. The reason being that, looking at the data altogether, one could reasonably posit that it is actually increasing NREMS more so than decreasing REMS driving the reduction in the REMS % TST value after higher doses of CNO at least for the acute phase. But that could be left to the interpretation of the reader.

Overall, the authors adequately addressed the majority of my concerns, but a few remain as detailed below:

1) I appreciate the authors detailed response for multiple testing considerations and for now including those analyses. However, the rationale for not conducting an ANOVA for the EEG spectral data prior to (what is typically post-hoc) pairwise testing is lacking. The authors cite Achermann and Borbély (1998) in their response to support not conducting an ANOVA; however, Achermann and Borbély state that they conducted pairwise testing post-hoc only if the ANOVA for repeated measures was significant. This testing occurred over 198 frequency bins. Using an ANOVA prior to pairwise testing may also prevent unnecessary pairwise testing that would reveal a few significant scattered spectral bins that do not reflect a systematic change that the authors detail in the remainder of their response which focuses on multiple testing corrections for pairwise tests and bin size considerations.

2) Related, the authors refer to the pairwise testing of EEG spectral data as "post-hoc" testing, but statistically speaking, none of these comparisons are "post-hoc" because ANOVAs were not conducted prior to pairwise testing. See Figure 3, Figure 5, Figure 3 —figure supplement 1, Figure 3 —figure supplement 2, Figure 3 —figure supplement 3, and Figure 5 —figure supplement 1.

3) To improve clarity, it would be helpful to edit Page 6, Line 125 and Page 20, Line 345 to "suppression of REM sleep relative to NREM sleep." Similarly, editing Figure 3e from "REM sleep" to "REM sleep (%TST)" or REM/NREM sleep" would also help.

4) On Page 12, Lines 215 – 216, the authors state that less REM sleep is typically observed during recovery from sleep deprivation. This is not typically the case, and the cited supporting text (Huber, Deboer, and Tobler 2000) shows that REMS either increases or does not differ after sleep deprivation compared to baseline.

5) The authors state in the Figure 1 —figure supplement 1 caption, "Note the relatively high variability in the percentage of REM sleep as percentage of total recording time between animals (bottom row panels)…" (Page 36, Lines 704 – 705). This statement is not supported by the data shown. The individual variability and error bars are similar across all analyses displayed in these plots. Also, the SEM values listed in Supplementary Table 1 are lower in all conditions calculated as %TRT compared to %TST, and the SEM values are of similar magnitude relative to their respective means for both data sets. This also applies to the statement on Page 8, Line 133 about high variability of sleep amounts between individual animals.

6) Figure 3 caption should say post-hoc comparisons are shown for panels c and d.

---

## [Author Response]

[Editors’ note: the authors resubmitted a revised version of the paper for consideration. What follows is the authors’ response to the first round of review.]

Reviewer #1 (Recommendations for the authors):Traut, Mengual et al. investigated the impact of the chemogenetic ligands clozapine N-oxide (CNO) and compound 21 (C21) on sleep in wildtype C57Bl/6J mice. They report that intraperitoneal injection of three CNO doses (1, 5, and 10 mg/kg) dose-dependently suppresses rapid eye movement (REM) sleep, consolidates non-rapid eye movement (NREMS), and alters NREM electroencephalographic (EEG) spectra based on standard EEG/electromyographic (EMG) assessment of sleep-wake behavior. The sleep field has generally attributed CNO effects on sleep to back-metabolism of CNO to sleep-promoting clozapine; however, the authors also report similar sleep effects by C21 administration-a chemogenetic ligand that does not back-metabolize to clozapine. Therefore, the authors reason that the impact of chemogenetic ligands on sleep could also be due to off-target effects on endogenous neurotransmitter systems. This paper highlights the need for important non-DREADD expressing controls in sleep research, but this conclusion can be extended to other neuroscience research using chemogenetics. This conclusion is supported by the data using techniques that are easily implemented by those in the sleep field, but there are some aspects of analysis that can be improved.

We thank reviewer #1 for the positive assessment of our manuscript and the helpful suggestions regarding improvements to the data analysis.

1) The authors report post-hoc significance for statistical tests that did not have significant main effects (Figures 1b, 2c). In addition, no corrections for multiple testing of spectral data are implemented. The authors are transparent about this reporting, however.

We thank the reviewer for highlighting two important statistical considerations (I) the report of post-hoc significance for statistical tests that do not have significant main effects and (II) the question if correction for multiple testing of spectral data should be implemented.

(I) Report of post-hoc significance for statistical tests that do not have significant main effects We agree with the reviewer that post-hoc significance testing should generally only be presented for statistical tests that have significant main effects and we have corrected this issue in the updated manuscript.

We initially chose to present the post-hoc tests in Figures 1b and 2c as the main effects approach significance (Figure 1b: *p* = 0.0682 for percentage of NREM sleep in the first 2h following injections; Figure 2c: *p* = 0.0690 for number of REM episodes per hour; mixedeffect analysis) and because the post-hoc contrasts for the 10 mg/kg CNO condition had medium-to-large effect sizes (Cohen’s *d* = 0.6961 for percentage of NREM sleep; *d* = -0.9286 for number of REM episodes per hour) indicating potentially relevant effects. To be fully transparent about this non-significant result we had included a note in the figure legends that there is no main effect and therefore, the significant post hoc-contrasts in these two analyses should be interpreted with caution.

Following the suggestions of both reviewers, we have now removed the indicators of significant post-hoc tests from Figure 1b. We would also like to mention that the adjustment of the definition of REM episode duration (minimum duration 16 seconds instead of 1 minute) suggested by reviewer #1, has revealed a statistically significant main effect for the number of REM episodes shown in Figure 2c.

(II) Correction for multiple testing of spectral data

The presentation of absolute (not normalised) log-transformed EEG power spectra at a finescale resolution (e.g. 0.25 Hz bins) and with uncorrected pairwise comparison of individual spectral bins between groups represents the most unbiased approach to analyse EEG spectra in mice and can therefore be considered the gold standard approach.

The Benjamini-Hochberg (BH) procedure can easily be implemented but has some important limitations in its application to EEG spectral analysis as we briefly outline below. However, we respect the reviewer’s interest in finding out whether correction for multiple testing would change the conclusions of the spectral analysis. Our original results with bins reaching the uncorrected α error threshold of *p* = 0.05 now also include three additional plots highlighting the significant bins for (1) the more rigorous α error threshold of *p* = 0.01, (2) the BH procedure and (3) the BH procedure applied to 1 Hz instead of 0.25 Hz bins as suggested by the reviewer. This analysis highlights that systematic effects on EEG power spectra of NREM sleep are still present when corrections for multiple testing are applied. We now include the BH correction for all spectra in the supplementary figures. We thank the reviewer for suggesting this analysis and helping us to demonstrate the robustness of our finding that CNO systematically alters the NREM sleep spectrogram also after correction for multiple testing.

There are two main reasons why we and others do not perform ANOVAS, correction for multiple testing and/or reduction of the resolution (to 1 Hz bins or predefined power bands) in EEG spectral analysis as suggested by reviewer #1 below. The two reasons are as follows:

(1) Adjacent frequency bins do not vary independently, so a strict correction such as the Bonferroni method would be too conservative (Achermann and Borbély 1998). Due to the large number of bins (119 bins of 0.25 Hz in our case) the sensitivity to detect spectral differences is largely reduced, even when less conservative correction methods such as the suggested Benjamini-Hochberg (BH) procedure are applied. The reduced sensitivity due to overcorrection is of particular importance, when the BH procedure is applied to analyses where a large number of spectral bins show strong group differences. This is because the α error threshold for each individual analysis (i.e. frequency bin in the case of EEG spectral analysis) depends on the number of positives that rejected the null hypothesis before correction (Jafari and Ansari-Pour 2019; Benjamini and Hochberg 1995). Therefore, when applying the BH procedure, a given difference in specific frequency bins might be detected as “statistically significant” or not, depending on the magnitude of differences in other frequencies. This makes it very difficult to compare results across experiments and studies.

This issue with the BH procedure can be demonstrated in our data by looking at the uncorrected and BH-corrected wake, NREM and REM spectrograms. In the wake and REM spectrograms, only a few scattered spectral bins are below the uncorrected α error threshold of *p* = 0.05. While it becomes immediately clear that these scattered bins do not reflect a systematic change in the EEG spectrogram and the resulting p-value of pairwise comparisons is >0.01, some of these bins remain significant even after correction with the BH approach due to the small initial number of bins rejecting the null hypothesis. In contrast, the systematic group difference in the adjacent frequency bins between 0.5 and 1.25 Hz of the NREM spectrogram which reach the conservative α error threshold of p = 0.01 do not become significant after BH correction due to the large number of bins with even stronger differences in the higher frequencies.

(2) Reducing the bin size or grouping spectral data into predefined power bands, the precise information about the boundaries of spectral differences is compromised. The historically defined EEG power bands vary between species and their boundaries do not adequately reflect the current knowledge on the oscillations generated by brain circuits (Adamantidis, Gutierrez Herrera, and Gent 2019). For example, subdivisions of δ-activity likely reflect different physiological processes during sleep (Hubbard et al. 2020). It must be highlighted that a high resolution (large number of small frequency bins) and wide frequency range (covering the frequency spectra of typical sleep oscillations) is desirable in EEG spectral power analysis. We demonstrate in Author response image 1 that reducing the resolution from 0.25 into 1 Hz bins followed by the BH approach does not improve the analysis.

The most unbiased approach to compare EEG spectra at a high resolution over a wide range is therefore to report the uncorrected t-tests for bin-wise comparisons. While individual ‘significant’ bins are uninformative, several neighbouring ‘significant’ bins allow one to identify ranges of systematic power differences. To inform the reader in detail about the statistical robustness of differences at individual bins, we now provide four significance plots (p<0.05, p<0.01 and BH-correction with 0.25 and 1 Hz resolution) under each spectrogram.

**Author response image 1. sa2fig1:** Benjamini-Hochberg correction for multiple comparisons confirms the robustness of CNO effects on NREM sleep EEG spectra. Note that the stringent α error threshold of p<0.01 (black asterisks) largely reflects the result of the uncorrected t-tests. Note that the Benjamini-Hochberg correction (light blue asterisks) fails to remove scattered positive bins that do not reach the α error threshold of p<0.01 when few bins are positive (e.g. Wake and REM spectra in the time window 0-2 hours) but removes potentially biologically relevant ranges of significant differences when many individual bins show stronger group differences (see spectral range <1.25 Hz in NREM spectra in the time window 0-2 hours). Reducing the resolution before applying BH correction can fail to detect small spectral ranges with strong difference but can highlight spectral ranges with several slight difference (see REM spectra in the time window 0-6 hours).

2) Analyses of REM sleep can be improved. Time spent in REM sleep is reported as a percentage of total sleep time which means that REM sleep is being assessed using different comparators across vehicle and CNO doses. Additionally, REM sleep architecture is based on REM sleep episodes {greater than or equal to} 1 min which tends to be longer than an average REM sleep episode in a mouse.

We thank the reviewer for these excellent suggestions which we have now implemented. The suggested analyses have provided us with a more nuanced understanding of the REM sleep suppression. The reduction in the minimum duration defining a REM episode further revealed significant effects of CNO and C21 on the number of REM sleep episodes. We now also found a significant increase in the REM latency for the C21 condition. In our initial submission, we had failed to detect these significant effects on REM sleep regulation with due to suboptimal parameters for REM sleep analysis.

The reviewer mentions two important issues regarding REM sleep analysis: (I) which comparator should be used to analyse the percentage of time spent in REM sleep and (II) which minimum duration defines a REM sleep episode in mice.

(I) comparator for time spent in REM sleep

Physiologically, REM sleep is entered through NREM sleep. Therefore, the timing and overall amount of REM sleep depend on the onset and the total amount of NREM sleep. For this reason, if NREM onset or NREM amount are altered, the most accurate measure for the amount of REM sleep is to analyse REM sleep as percentage of the overall amount of sleep (Franken, Malafosse, and Tafti 1999; Huber, Deboer, and Tobler 1999; Kashiwagi et al. 2020). Another possibility is to analyse the REM/NREM ratio. Both analyses allow to assess whether REM sleep regulation is altered in an animal model even when sleep amount and sleep onset times are altered.

For a more detailed understanding of the REM dysregulation following CNO/C21 injections, we now repeated all REM sleep analyses with all three measures of REM sleep (1) REM as percentage of total sleep time, (2) the REM/NREM ratio, and (3) REM as percentage of total recording time. We have included the new analyses as figure supplements to Figures 1 and 4 and in the supplementary tables that detail the results of the statistical analyses.

For CNO, the main effects of REM sleep expressed as percentage of total recording time approach significance but do not reach the α error threshold of 0.05 (p=0.0839 for 2h time window, p=0.0601 for 6h time window), therefore no post-hoc significance testing was performed. It should, however, be mentioned that there were medium-to-large effect sizes for post-hoc comparisons in the high dose CNO condition (Cohen’s d for 10 mg/kg CNO condition in 2h time window: -0.6559, in 6h time window -0.8140). For C21, all three measures of REM sleep show a significant reduction in the 6h time window.

The rationale for expressing REM sleep as percentage of total sleep time and the wellknown dependency of REM sleep on NREM sleep can be explained with an intuitive example: Suppose we perform gentle handling with an animal one morning to keep it awake, this would not be called a “REM suppressing manipulation” despite the fact that the total amount of REM sleep in our six-hour recording period would be reduced. Similarly, if the same animal received a hypnotic the next morning and fell asleep immediately, no one would call this a “REM boosting effect” despite the fact that there would be more REM sleep during the recording period. Everyone would agree that the REM regulation in this animal is unaffected by these manipulations if the proportion of REM sleep relative to the overall amount of sleep and the REM/NREM ratio remains unchanged.

In conclusion, our analyses unequivocally show that REM sleep is suppressed following CNO and C21 injections. The lack of a significant main effect for CNO on the percentage of REM sleep per recording period demonstrates that changes in sleep architecture can mask the REM sleep suppression. The trend towards a larger overall amount of NREM sleep and shorter NREM sleep latencies following CNO injections explain why the effect sizes for the reduction in the percentage of REM sleep relative to the recording time are smaller compared to the REM/NREM ratio or the percentage of REM relative to the total sleep time. This detailed analysis explains why other studies might have missed the REM sleep suppressing effect of DREADD actuators by merely analysing the overall amount of REM sleep over a fixed time interval. We have now updated the text in the main manuscript to include these new findings and important considerations.

(II) minimum duration of REM sleep episodes

We would like to thank the reviewer for making us aware that our minimum duration of 1 min for the definition of REM sleep episodes was unusually long for mice and therefore many REM sleep episodes were missed in our initial sleep architectural analysis. We therefore previously failed to detect the significant differences in the number of REM episodes following CNO and C21 injections and in the REM latency following C21 injections.

We initially defined ‘vigilance state episodes’ as 1 minute (15 x 4s epochs) or longer spent in any vigilance state with a maximum of 16 seconds (4 x 4s epochs) interruption as we used this definition in a previous publication to define NREM and wake episodes (Krone et al. 2021). However, as the reviewer points out, this definition is not suited for the analysis of REM sleep episodes, which are often shorter than 1 min in C57BL/6 mice (Huber, Deboer, and Tobler 2000; McShane et al. 2010). Following the reviewer’s suggestion, we now repeated all analyses of individual REM episodes with a shorter cut-off (16s, i.e. 4 x 4 s epochs) and interruption criterion (8s, i.e. 2 x 4s epochs). These analyses confirmed our previous results on altered REM sleep architecture following CNO and C21 injections. However, the new analyses also revealed a significant effect of CNO and C21 on the number of REM episodes that we had previously missed. The reviewers’ comment also made us aware that due to the suboptimal definition of REM episodes we missed a significant effect of C21 on REM latency, which we had only seen in the CNO condition previously, as well as a significant effect of the 1 mg/kg CNO in addition to the 10 mg/kg CNO condition because we had calculated the latency to the first REM episode (of minimum 1 minute) rather than to the first REM epoch.

We would like to clarify that for the calculation of time spent in any vigilance state, all 4s epochs were already included in the initial analysis and therefore the proportion of time spent in vigilance states is unaffected by the parameters that define individual wake, NREM or REM sleep episodes. We would also like to point out that there is no gold standard for the parameters that define vigilance state episodes in rodents (Trachsel et al. 1991) and approaches such as state transition analysis can provide additional information on the robustness of sleep architectural changes (Banks et al. 2020). We have now performed such an analysis of the transition/continuation probability from one 4s-epoch to the next and find that 5 and 10 mg/kg CNO as well as 3 mg/kg C21 significantly increase the probability to remain in REM and NREM sleep once this state has been entered. This new finding of increased sleep state stability is in line with previous reports on increased ‘sleep maintenance’ following clozapine-injections in rats (Sorge, Pollmächer, and Lancel 2004).

In the updated manuscript, we have replaced the panels relating to REM sleep in figures 2, 3 and 5. We have further added the state transition analysis to figures 3 and 5 and updated the description of the results accordingly.

Once again, we would like to thank the reviewer for suggesting these additional analyses which helped us develop a more nuanced understanding of the REM suppressing effects of CNO and C21 and revealed some additional effects on REM sleep regulation which we had missed previously. We also provide a new sleep state analysis that shows increased sleep state stability following 5 and 10 mg/kg CNO and 3 mg/kg C21 injections.

3) Authors acknowledge that a limitation of this study is that many of the animals included were intended for different experimental purposes and underwent other procedures prior to this study. However, the within-subject design and rest periods prior to this study should reduce potential impacts of prior experimental procedures on the present findings.

We fully agree with the reviewer and thank them for highlighting the value of our within subject design. Following the suggestion from reviewer #2 we have also repeated all analyses with the 11 animals that had all CNO conditions and could demonstrate that the effects remain robust in a full within-subject design.

Recommendations1) Although I appreciate the authors' transparency, there are a few corrections that should be made in terms of statistical reporting:a. Do not round the degrees of freedom reported for the F statistic in the text.

We now report the unrounded values for the degrees of freedom for the *F* statistic in the text.

b. If there are no significant main effects, post-hoc analyses should not be initiated. Therefore, the post-hoc data reporting and significance markers should be removed for Figures 1b and 2c.

This issue has been corrected in the updated manuscript. The significance marker on Figure 1b has been removed. Following the reduction of the minimum duration that defines a REM episode, as suggested by this reviewer, Figure 2c (REM episode number) now shows a significant main effect and significant post-hoc analyses for the comparison of saline with 5 mg/kg and 10 mg/kg CNO.

c. It is not clear if a repeated measures ANOVA (or a similar test) was run prior to pairwise comparisons of spectral data. Also, these pairwise comparisons need to be corrected for multiple testing using something like a Benjamini-Hochberg correction. One option to help with spectral analyses is to reduce the resolution to 0.5 or 1 Hz without compromising the integrity of the data.

Please see detailed response above on spectral analysis. Following the request of the reviewer, we have now implemented the Benjamini-Hochberg (BH) correction both with and without reducing the resolution. These additional analyses show that the systematic effects of 5 and 10 mg/kg CNO and 3 mg/kg C21 on NREM sleep spectra are robust.

d. The authors describe post-hoc contrasts with regard to t-tests for C21 analyses.

This error in the legends of Figures 4 and 5 has been corrected.

2) What is the rationale for reporting time spent in REM sleep as a percentage of total sleep time? This analysis uses different comparators (total NREM + REM vs total recording time) across conditions. I think it would be better to report REM sleep as a percentage of total recording time in Figures 1 and 4 as is reported for wakefulness and NREM sleep to get a more direct assessment of REM sleep suppression. Reporting REM sleep as a percentage of total sleep time can still be useful for assessing NREM/REM ratios.

Please see the detailed reply on REM sleep analysis above. We now provide all three measurements (REM sleep as percentage of total recording time, REM sleep as percentage of total sleep time, and REM/NREM ratios) for a more detailed presentation of the effects of CNO and C21 on REM sleep.

3) Using {greater than or equal to}1 min to define a REM episode seems quite long for a mouse (see McShane et al., J Neurosci Methods, 2010). This criterion could be excluding a majority of REM episodes.

Please see detailed reply on REM sleep analysis above. We fully agree with the reviewer that the previous minimum duration for the definition of REM episodes was ususually long and have now corrected the defining criteria for REM episodes to reflect the considerably shorter average episode duration of REM episodes compared to wake or NREM episodes. The minimum duration is now 16 s (instead of 1 min) and the permitted interruption 8 s (instead of 16 s). The updated criteria have revealed additional effects of CNO and C21 on REM sleep architecture that we had previously missed.

4) Many of the figures contain both red and green which makes it difficult for some readers to differentiate between conditions, especially in Figures 1a, 2a, and 3a. Please switch to a more inclusive color palette. In addition, it is difficult to see the 5 mg/kg data in Figure 3a despite that data being the primary focus of the text.

We thank the reviewer for this important consideration. We have now implemented the *eLife* recommendations on Color Universal Design (CUD) to make our figures friendly to colour-blind people. While we maintain the colour coding green-orange-red to indicate increasing doses of CNO, we now use redundantly coded drawings (using symbols and line types in addition to colours to differentiate groups) to make all figures clearly legible for colourblind individuals. We now also display the three conditions in Figure 3a on separate graphs to improve the visibility of the 5 mg/kg group, which was barely visible in the previous figure as it almost entirely overlaps with the 10 mg/kg group. We have exported all figures in black/white design and had them checked by a colour-blind individual to ensure that none of the updated figure panels in the manuscript requires colour coding.

5) Page 8, Line 157 seems to state in error that "REM sleep episodes were significantly shorter…"

This error has been corrected in the updated manuscript.

6) Because sleep impacts a vast number of biological processes and behaviors, it would be helpful to briefly include in the Discussion how these findings can impact the broader neuroscience community beyond the sleep field. Also, it might be helpful to include what is known (if anything) on chemogenetic ligand administration during the dark period.

We thank the reviewer for highlighting that our results impact the broader neuroscience community beyond the sleep field. Due to the wide range of neurotransmitter systems involved in the regulation of sleep amount, sleep oscillations and sleep architecture, our study shows that sleep recordings represent an excellent tool to test the biological inertness of novel designer drugs in vivo. We were also made aware that our preprint was the first demonstration that DREADD actuators which cannot convert into clozapine can elicit behavioural effects in DREADD-free animals. As sleep impacts nearly any behaviour, in particular learning and memory, the sleep-modulating effects of DREADD actuators are of great relevance for anyone conducting neuroscience research using chemogenetics. As suggested by the reviewer, we have now included these important considerations in the updated Discussion section:

“While it has been described that C21 can affect neuronal firing (Goutardier, 2020), to our knowledge this is the first study demonstrating behavioural effects of a DREADD actuator that cannot convert to clozapine.”

and

“Due to the structural similarity between C21, perlapine and other novel chemogenetic actuators, such as deschloroclozapine (Nagai et al. 2020) and JHU37152 and JHU37160 (Bonaventura et al. 2019), systematically testing the impact of these actuators on sleep appears to be of paramount importance to validate their inertness in vivo.”

and

“Our work indicates that sleep analysis can reveal behavioural effects of DREADD agonists that are missed by other established behavioural tests such as the elevated plus maze and the marble burying task or measurements of locomotion and reaction-time (Jendryka et al. 2019; Tran et al. 2020). Considering the complexity of neurotransmitter systems regulating sleep (Saper and Fuller 2017) and the sensitivity of sleep architecture to pharmacological intervention (Riemann and Nissen 2012), we propose that sleep assessment could serve as an invaluable tool to evaluate the biological inertness of newly developed chemogenetic actuators.”

We injected the DREADD actuators at light onset because the early light period is the zeitgeber time at which animals typically sleep. Therefore, sleep-modulating effects of drugs are typically tested by application at the beginning of the light period (Kopp et al. 2002; Sorge, Pollmächer, and Lancel 2004; McKillop et al. 2021; Thomas et al. 2022). In contrast, the dark period is the mice’s active period and little REM and NREM sleep occurs (Franken, Malafosse, and Tafti 1999; Huber, Deboer, and Tobler 2000). One could speculate that we might have found a significant increase in NREM sleep (instead of a non-significant trend) if we had injected CNO and C21 during the dark period when the baseline level of NREM is low. This would be in line with previous reports of increased NREM sleep after injections of clozapine in rats (Sorge, Pollmächer, and Lancel 2004) and CNO in mice (Varin, Luppi, and Fort 2018). In contrast, the low amount of sleep during the dark period might overall hamper the detection of sleep-modulating effects. This might be one of the reasons why a recent study found no significant effects on a small set of assessed sleep parameters after intraperitoneal injections of DREADD actuators at dark onset (Ferrari et al. 2022). Following the reviewer’s suggestion, we now discuss potential differences in the effects of chemogenetic ligand administration on sleep between light and dark period in the Discussion section of the manuscript:

“In addition, differences in the zeitgeber time of drug application need to be considered when comparing our results to other studies. While sleep-modulating effects of drugs are usually assessed by injecting at light onset and assessing sleep during the early light period (Kopp et al. 2002; Sorge, Pollmächer, and Lancel 2004; McKillop et al. 2021; Thomas et al. 2022), the time of day when mice sleep most, sleep-modulating effects of DREADD actuators might be less pronounced when drugs are injected at dark onset and sleep is assessed during the early dark period when mice are typically active and sleep little (Ferrari et al. 2022).”

Reviewer #2 (Recommendations for the authors):The use of DREADDs in sleep research is common. In typical experiments, it is assumed that the DREADD actuators on their own do not affect sleep, and a typical control group may receive saline i.p. injections that are considered adequate to control for the effects of injection/animal handling on sleep. However, here the authors show that intraperitoneal application of the common DREADD actuators Clozapine-N-oxide (CNO) and compound 21 (C21) modulate sleep architecture and sleep EEG features compared to saline. Specifically, CNO reduced REM sleep, led to longer bouts of NREM sleep and elevated EEG power in the lower frequencies at the expense of power > 6Hz.The experiments and analysis are carefully conducted and the manuscript is clearly written. The results are nicely put in the context of existing literature and their implications are well discussed. In addition, the use of C21 elegantly addresses the issue of clozapine back-metabolism. At the same time, it is evident that the authors started this investigation as a control experiment for another project. As such, the experimental design and data acquisition protocols are suboptimal to address the question at hand. For example, different experiments and analyses have a different number of animals, and conditions are not necessarily counterbalanced in their order. This diminishes statistical power (since comparisons are performed between a limited group of subjects, rather than within subjects). While this work is important for the research community, it is quite technical and the overall impression is that results are modest and may not be very robust.While this work is important for the research community, my enthusiasm is dampened by two issues:

We thank reviewer #2 for the largely positive assessment of our manuscript and the helpful suggestions to probe the robustness of our findings. We could not agree more that our work is *“quite technical”*. However, in contrast to the reviewer, we consider this a particular strength of this manuscript. As both reviewers highlight in their summaries, our findings that DREADD-actuators, independent of back-metabolism to clozapine, can influence behaviour of DREADD-free animals is a novel and important finding for the design of future chemogenetic experiments in all fields of neuroscience and will therefore *“impact the broader neuroscience community beyond the sleep field”* as pointed out by reviewer #1. While our initial manuscript had some shortcomings in the statistical analysis and data presentation, the updated analysis and additional data addresses these shortcomings and leaves no doubt that several sleep parameters are strongly and robustly altered by CNO and C21.

1. The overall impression is that the results are modest and not entirely robust. Different effects are observed for different drug doses and different features of sleep (e.g. NREM is longer but with fewer bouts, but no effect on overall % of NREM; some effects such as delayed transitions to REM sleep are strongest with low doses; EEG spectrogram effects are most evident at middle doses). Given the many tests employed (NREM/REM * different features such as number of bouts or their duration * many temporal intervals examined separately * 3 doses) there may be a multiple comparison challenge that, together with the limited number of animals in some experiments (n=3 EEG spectral analysis in C21), and some statistical issues described below, raise some doubts about whether robust conclusions can be drawn.

The reviewer raises a number of important considerations that deserve to be discussed individually:

(I) “Results are modest and not entirely robust”

With n=15 animals, our within-subject and counterbalanced comparison of the 5 mg/kg CNO condition with saline presents to our knowledge the largest and most robust comparison of sleep parameters between DREADD actuator and vehicle in DREADDfree mice. Control groups in sleep experiments using DREADDs are typically smaller, ranging from n=3-6 (Funk et al. 2017), to n=10 (Varin, Luppi, and Fort 2018), or n=14 (Harding et al. 2018) and often do not include a within-subject design. For example a recent study including a dose-dependent control experiment for CNO and C21 compared four conditions in n=7 animals and comes to the conclusion that “these agonists do not on their own interfere with sleep in control (not expressing DREADDs) mice” (Figure 3 panel C5 from Ferrari et al. 2022). We would like to highlight that previous studies which came to the conclusion that DREADD actuators do not affect sleep typically reported only a limited set of sleep parameters and had smaller sample sizes as well as less statistical power than our study. This might have led to the common problem of over-interpreting nonsignificant results (Makin and Orban de Xivry 2019). In addition to reporting the results from null-hypothesis testing, we provide raw data and effect sizes for a comprehensive set of sleep metrics to allow anyone to verify the strength and robustness of our results.

For several sleep parameters, especially sleep architecture and sleep consolidation measures, we find effect sizes of Cohen’s d >0.8 for the 5 mg/kg CNO group, but also for the 10 mg/kg CNO and 3 mg/kg C21 groups, indicating large to very large effects for the respective measures and CNO/C21 doses.

While we agree with the reviewer that the effect of CNO and C21 on the overall amount of sleep could be considered modest – and therefore sleep-modulating effects of these actuators have often been missed in the past – our results that both DREADD actuators, CNO and C21, affect the proportion of REM sleep, sleep architecture, EEG power spectra of NREM sleep, sleep stability and continuity are strong and robust.

(II) Different effects are observed for different drug doses and different features of sleep

“NREM is longer but with fewer bouts, but no effect on overall %”

It was previously described in rats that clozapine injections (Sorge, Pollmächer, and Lancel 2004), lead to a robust, strong, and dose-dependent increase in NREM episode duration and a decrease in NREM episode number. However, both low and high doses of clozapine caused only a moderate increase in the overall percentage of NREM sleep and high-dose clozapine injections even caused an initial reduction in the percentage of NREM sleep for the first 2 hours followed by an increase in NREM sleep (Sorge, Pollmächer, and Lancel 2004). The stronger effects of clozapine-related substances on NREM sleep architecture compared to the effects on NREM sleep percentage is reflected in our effect size analysis for CNO and C21 in supplementary tables 1-3 of the manuscript. In summary, our results of CNO and C21 effects on the architecture and the overall percentage of NREM sleep in mice are strikingly consistent with previously described effects of clozapine injections in rats. This underlines the biological plausibility of our results.

“some effects such as delayed transitions to REM sleep are strongest with low doses”

Nearly all CNO effects follow a dose-dependent pattern, yet for two sleep architectural parameters, the NREM latency and REM latency, the effects appear to be strongest for the lowest CNO dose (1 mg/kg). For clozapine-injections in rats, it has been described that small doses of clozapine (2.5 mg/kg) cause an immediate small and short-lasting increase in NREM sleep, while high doses of clozapine (7.5 mg/kg) cause an initial (paradoxical) increase in wakefulness before leading to a sustained increase in NREM sleep (Sorge, Pollmächer, and Lancel 2004). It could therefore be considered that potentially the low dose of 1 mg/kg CNO affects the latency to NREM and REM sleep more strongly than higher doses. However, we would like to use this opportunity to highlight in full transparency, as already outlined in the previous manuscript, that sequence effects for the 1 mg/kg and 10 mg/kg CNO condition are possible (i.e. all animals having received 5 mg/kg CNO and saline injections prior to the 1 mg/kg and 10 mg/kg conditions in order to avoid potential habituation to the designer drug). One could speculate that the NREM/REM onset latencies might be the sleep measures that are most sensitive to handling or stress as a confounder, and therefore the pharmacological effects were most strongly visible in the conditions (1 mg/kg and 10 mg/kg) where the animals were already habituated to the injection procedure. While we cannot fully exclude sequence effects for the 1 mg/kg and 10 mg/kg CNO conditions, the dose-dependency of the vast majority of CNO effects on sleep variables indicates a genuine pharmacological effect.

EEG spectrogram effects are most evident at middle doses

This statement is not correct and must be due to a misunderstanding. To clarify our results: we find no systematic effects of 1 mg/kg CNO on EEG spectra; the spectral changes in the NREM spectrogram after 5 mg/kg and 10 mg/kg CNO injections are nearly identical. The effects of 10 mg/kg CNO on slow frequencies do not reach significance, most likely due to the smaller sample size of the 10 mg/kg CNO condition (n=8) compared to the 5 mg/kg condition (n=10).

We have now included the absolute spectra for all vigilance states and CNO doses as figure supplements in the main manuscript and include them for the benefit of the reviewers (Author response image 2; Author response image 3; Author response image 4) to avoid any further misunderstandings of the data. We have also revised the wording in the Results section of the main manuscript to clarify this important point.

**Author response image 2. sa2fig2:** Frontal EEG spectra in comparison between 1 mg/kg CNO and saline injections.

**Author response image 3. sa2fig3:** Frontal EEG spectra in comparison between 5 mg/kg CNO and saline injections.

**Author response image 4. sa2fig4:** Frontal EEG spectra in comparison between 10 mg/kg CNO and saline injections.

(III) Given the many tests employed (NREM/REM * different features such as number of bouts or their duration * many temporal intervals examined separately * 3 doses) there may be a multiple comparison challenge that, together with the limited number of animals in some experiments (n=3 EEG spectral analysis in C21), and some statistical issues described below, raise some doubts about whether robust conclusions can be drawn.

As mentioned above, our sample size is larger and the within-subject design and counterbalancing of the 5 mg/kg CNO with the saline condition more rigorous than that of previous experiments that claim the absence of effects of DREADD actuators on sleep in DREADD-free animals. The reviewer is correct that multiple comparisons pose a challenge to any experiments where several measures are examined. However, it needs to be emphasised that the decision on when and how to correct of multiple testing has to be made with care (Jafari and Ansari-Pour 2019) because conservative approaches to correct for multiple testing can lead to a substantial reduction in statistical power and, thereby, biologically relevant effects can be missed by over-interpreting non-significant results (Benjamini and Hochberg 1995; Makin and Orban de Xivry 2019)**.**

We would like to mention that we have now performed BH-corrections for multiple testing for the spectral analysis as suggested by reviewer #1 that demonstrate the robustness of our EEG spectral results.

Regarding vigilance time, sleep architecture and sleep consolidation measures, we performed 19 analyses for the CNO experiment, 10 of which indicated significant effects of the drug injections. The number of false positives expected by chance for an α error probability of *p* = 0.05, however, would be ~1 (λ = 0.05*19=0.95). Using a Poisson distribution, we calculated the likelihood that this result occurred by chance and found this to be 6.38*10^-8^. In addition and in line with recent recommendations on corrections for multiple testing (Jafari and Ansari-Pour 2019; Makin and Orban de Xivry 2019), we present the values of individual animals and effect size measurements, indicating medium, large, or very large effects for several of the analyses for which the null-hypothesis was rejected.

It should further be mentioned that our effects are biologically plausible, given the striking similarity of our CNO and C21 effects with the long-established sleep modulating properties of the structurally similar atypical antipsychotic clozapine in rodents (Sorge, Pollmächer, and Lancel 2004; Spierings, Dzoljic, and Godschalk 1977).

The reviewer highlights that one analysis, EEG spectra after C21 injections, was based on a sample size of n=3, which was due to technical issues with the implantation of the EEG headstage leading to EEG artefacts in one batch of animals. We agree with the reviewer that for this sample size no null-hypothesis testing should have been performed. We carefully double-checked the spectrograms of individual animals and found that our initial exclusion of 4 out of 7 C21-injected animals from the spectral analysis was too conservative, as we excluded 3 of these animals from the spectral analysis of all vigilance states merely because of movement artefacts in the wake state. We were now able to repeat the EEG spectral analysis of NREM sleep for n=6 instead of n=3 mice injected with C21 and found the results to be statistically significant, robust after correction for multiple testing, and highly consistent with those of the 5 and 10 mg/kg CNO conditions (Author response image 5).

**Author response image 5. sa2fig5:** Absolute and relative frontal EEG spectra of NREM sleep in comparison between 3 mg/kg C21 and saline injections in n=6 animals. a) absolute and b) relative EEG spectra. Note that this analysis performed on n=6 animals with artefact-free NREM sleep resembles the EEG spectral changes observed in the 5 and 10 mg/kg CNO condition.

All of our statistical analyses in the updated manuscript are now based on at least n=6 animals, which is as large or larger than the DREADD-free control group in several studies claiming no significant effects of DREADD actuators on sleep. Following the additional analyses suggested by the reviewers, our data leaves no doubt that the finding of sleep-modulating effects of CNO and C21 is robust. We now provide the raw data and the analysis scripts in an online repository, which allows anyone to test their robustness and reproduce our results.

2. The other issue is that I find this report very technical. Although important for investigators in the field, it may not appeal to a wide audience.With the combination of these two issues, I don't think that the manuscript in its present form is suitable for eLife.

We thank the reviewer for acknowledging that our work is *“important for investigators in the field”* and fully agree that this report is *“very technical”*. We consider this to be a strength, not a weakness. Chemogenetics has become a leading technique in sleep research (155 Pubmed-listed articles on ‘chemogen*’ AND ‘sleep’, 07^th^ October 2022) and neuroscience (1817 Pubmed-listed articles on ‘chemogen*’ AND ‘neuro*’, 07^th^ October 2022) with potential clinical applications on the horizon (Singer et al. 2022; Fleury Curado et al. 2018; Venner et al. 2019). As reviewer #1 pointed out, our finding that chemogenetic actuators that do not convert to clozapine can still evoke clozapine-like effects in vivo, potentially due to off-target receptor binding, has wide-ranging implications for the future implementation of chemogenetics (see detailed comment above). In addition, electrophysiological sleep assessment could be established as a simple but highly sensitive in vivo screening tool for the biological inertness of newly developed chemogenetic actuators due to the host of neurotransmitter systems involved in the regulation of sleep.

Supporting the broad interest in our research, we’re pleased to report that our submission has been accessed >3,400 times on BioRxiv since it appeared online in February 2022 and has attracted considerable attention on social media among the neuroscience community and reached a high Altmetric attention score (93^rd^ percentile compared to outputs of the same age). In addition, an abstract on the work was selected for presentation at three premier neuroscience and sleep research meetings: SFN 2021, the World Sleep Conference 2022, EuroSleep 2022. This project has been selected among several hundred submissions for a Young Scientists Award by the European Sleep Research Society. While these are good indicators that our research appeals to a wide audience, we would like to highlight that according to the San Francisco Declaration on Research Assessment (https://sfdora.org/read/), of which *eLife* was a founder and continues to be a supporter, a more essential impact measure is the influence on research practice. Our findings will undoubtedly impact the research practice in the application of DREADD technology, help explain and understand occasional inconsistent findings in previous chemogenetic experiments, and improve the quality of future scientific work applying chemogenetic tools. In addition, our openly accessible data set and collection of scripts will allow other researchers to replicate our findings as well as explore our and their own data for further effects of CNO and C21.

We are confident that the reviewer will appreciate that our BioRxiv preprint has already appealed to a wide audience and will be of great relevance for the broad audience of *eLife*.

Specific comments:– To be best of my knowledge, it may not be appropriate to perform Post-hoc statistical tests if initial ANOVA is not significant. Many other groups/papers would consider these results non-significant and not report them.

We agree with the reviewer and refer to our comment to reviewer #1 on this topic: In short, we have now corrected this issue and removed the presentation of the post-hoc statistical test in Figures 1b where the main analysis was not significant. The main effect of the mixed effect model for the number of REM episodes presented in Figure 2c did not reach, but only approached, statistical significance in our initial analysis using a rather long minimum duration for REM sleep episodes as pointed out by reviewer #1. Following the reduction in the cut-off time for the definition of REM episodes (from 1 minute to 16 seconds) to include more REM episodes as suggested by reviewer #1 we now find a significant main effect for the number of REM episodes and two significant post-hoc tests for 5 and 10 mg/kg CNO compared to saline. The updated analysis is now presented in Figure 2c.

– I wonder how the results would seem if authors were to use only the 11 mice that had all 4 conditions? I understand that power calculations indicate that 12-13 mice are necessary, but using a within-subjects approach could serve as a complementary approach with potential to yield more robust findings.

We thank the reviewer for this excellent suggestion. We have now repeated our analysis with the 11 mice that had all 4 conditions. The analysis of n=11 mice clearly shows that the effects of the main analysis were not driven by the 4 mice for which individual conditions were not performed or the 1 mouse for which two conditions could not be analysed due to technical issues. In contrast effect sizes are similar, and in some cases stronger, than for the full dataset.

We would also like to emphasise that the analysis of absolute EEG spectra was already performed in a pure within-subject design, comparing saline and the respective dose of the actuator for those animals that received both respective conditions.

Although we already provide raw data and extensive supplementary data tables, we would like to include this analysis at the discretion of the editors as it underlines the robustness of our findings.

**Author response table 1. sa2table1:** Comparison of CNO effects for all animals (n=16) and for the subgroup of animals (n=11) that received all four injections (saline and 3 doses of CNO). Note that despite the considerably reduced power, all but two previously significant analysis still approach significance (p < 0.1). Further note that the effect sizes are very similar between n=11 and n=16 animals, indicating that the effects of the main analysis were not driven by the 5 animals that did not undergo all three CNO treatment conditions.

	null-hypothesistesting (mixedeffect models)p-value	effect sizes1 mg/kg CNO	effect sizes5 mg/kg CNO	effect sizes10 mg/kg CNO				
Analysis								
	n=16	n=11	n=16	n=11	n=16	n=11	n=16	n=11
Wake (%, 0-2 h)	0.2504	0.4114	-0.0234	-0.0489	-0.3393	-0.3890	-0.4815	-0.3793
Wake (%, 0-6 h)	0.4928	0.5702	0.1467	0.1059	-0.1715	-0.2587	-0.2633	-0.2145
NREM (%, 0-2 h)	0.0682	0.2085	0.1157	0.1389	0.4941	0.4915	0.6961	0.5389
NREM (%, 0-6 h)	0.1661	0.3039	-0.0361	-0.0416	0.4130	0.4214	0.4630	0.3328
REM (% of TST, 0-2 h)	0.0019	0.0211	-0.5802	-0.6207	-0.5594	-0.4037	-1.2743	-1.3152
REM (% of TST, 0-6 h)	0.0038	0.0863	-0.4697	-0.3743	-0.7724	-0.5301	-0.9333	-0.8488
Longest NREM ep.	0.0001	0.0005	0.3562	0.3616	0.9438	0.9599	1.5129	1.2811
Av. NREM ep. Dur.	0.0001	0.0028	0.1810	0.0356	1.1430	1.1181	1.1883	0.9655
Number of NREM ep.	0.0010	0.0181	-0.2938	-0.1339	-1.0628	-1.0981	-1.1989	-1.0248
NREM latency	0.0463	0.1068	-1.2021	-1.1337	-0.3744	-0.4198	-0.2424	-0.1689
Longest REM ep.	0.6266	0.7213	-0.0083	-0.2382	0.1710	0.0684	0.2450	0.0163
Av. REM ep.	0.0232	0.0631	-0.3757	-0.5302	0.5463	0.4006	0.4111	0.1842
Number of REM ep.	0.0047	0.0741	-0.0892	0.0870	-0.7456	-0.5453	-0.8710	-0.5829
REM latency	0.0220	0.0532	0.8061	0.7955	0.3202	0.3279	0.9213	0.7851
Brief awakenings	0.0006	0.0106	-0.6271	-0.4125	-1.0691	-0.8290	-1.5760	-1.4014
NREM before REM	0.0137	0.0577	0.7996	0.8960	0.5317	0.6101	1.8239	1.6994

– C 21 data: sample seems too small (especially for the spectral analysis with only n=3 mice) and lack dose dependency analysis. This makes it difficult to compare effect sizes for CNO and C21.

The C21 dataset has the standard sample size of n=6-7, typically used for electrophysiological sleep analysis (Gent et al. 2018) and our sample size is equal to or larger than that of the DREADD-free control groups in several chemogenetic experiments that claim no effects of DREADD actuators on sleep. For comparison, Ferrari et al. used 6-7 mice in an ANOVA analysis with 4 conditions to come to the conclusion that DREADD actuators have no effects on sleep in DREADD-free mice (Ferrari et al. 2022), Funk et al. used 3-6 DREADD-free mice as control group (Funk et al. 2017), Hayashi et al. used 48 mice in chemogenetic experiments testing the sleep-modulating features of specific neuronal populations (Hayashi et al. 2015).

We agree with the reviewer that with n=3 no null-hypothesis testing for EEG spectra in the C21 condition should have been performed. As mentioned above, in three of the four C21injected animals initially excluded from all EEG spectral analysis, the reason for exclusion were movement artefacts in the wake state. We were now able to include the EEG data from n=6 C21-injected animals for spectral analysis of NREM sleep and observed a moderate increase in low-frequency power and a significant and robust (BH-corrected for multiple testing) suppression of higher frequencies – strikingly similar to the changes to the NREM spectrogram observed in the 5 and 10 mg/kg CNO condition.

We included this data in an updated figure panel (Figure 5c) and in more detail in two figure supplements (Figure 5 —figure supplements 1 and 2). We removed the indication of null hypothesis testing from the wake and REM sleep spectra in Figure 5 —figure supplements 2 that are based on n=3 and would be happy to omit this supplementary figure at the discretion of the reviewers and editors.

– There are places where statistical approach may suffer from a multiple comparison issue. For example, in figure 1 the significance threshold is set at 0.05 but the % in each state (wake/NREM/REM) is evaluated for each dose/condition, and for multiple temporal intervals (six 1h bins). Consequently, the minimal p value may have to be adjusted.

Our presentation of the statistical approaches and results for the analysis of the proportion of time spent in vigilance states might have been suboptimal and therefore misunderstood. Exactly due to the potential issue with multiple comparisons highlighted by the reviewer, we did not analyse each 1h bin for each dose/condition and state. Instead, the proportion of time spent in vigilance states was analysed for acute (0-2 h) and prolonged (06 h) time windows. We used mixed effect models followed by Dunnett’s adjustment for posthoc comparisons. Even when conservative Bonferroni-correction is applied (six analyses, *p*_adjusted_=0.0083), the suppression of REM sleep remains significant. We have now updated the presentation of the statistical methods and results to avoid any misunderstanding. We would like to underline that the reporting of effect sizes can overcome some of the issues with null-hypothesis testing such as multiple corrections or black/white interpretation of results that reach a given significance threshold (Makin and Orban de Xivry 2019). Our post-hoc analyses indicate medium-to strong effects even for the percentage of time spent in NREM and REM sleep for the 5 mg/kg and 10 mg/kg conditions. The effect sizes for several of the sleep architecture and sleep continuity analyses are even stronger.

– Line 157: "REM sleep episodes were significantly shorter following CNO injections" but in Figure 2C second panel from right it seems that REM episodes are longer.

This error has been corrected.

– While most results are dose-dependent latency effects are strongest for lowest dose, which seems strange, even though the authors provide some thoughts in the discussion.

Please see our detailed answer on this topic above.

– Figure 2- the color green is both wake (in panel a) and 1 mg/kg (throughout the paper). To avoid confusion, it could be best to select different colors.

We have now implemented the *eLife* recommendations on Color Universal Design (CUD) for all figures (see detailed response to reviewer #1 above). The colour schemes have been improved. We now present the hypnograms in an updated colour scheme, optimised for colourblind individuals and with colours clearly distinguishable from that of the drug doses.

– In line 184 it is mentioned that the comparison was between 5 mg/kg and saline. You show all 3 doses in the main figure and in the supplementary you show all 3 states and not only NREM as described in text.

We would like to clarify that *“the focus of the EEG spectral analysis was on the comparison between the medium dose (5 mg/kg) of CNO and saline during NREM sleep”*, as stated in the previous version of the manuscript, because these were the two counterbalanced conditions and EEG spectra of NREM sleep are most commonly included in sleep publications due to the interest in slow oscillations and sleep spindles that occur in this sleep stage. For full transparency, we also included the other sleep stages and CNO doses although this was not the main focus of our analysis.

Following a request by reviewer #1, we have now added figure supplements with spectral analysis across all CNO doses and all vigilance states. The focus, however should remain on the comparison of the 5 mg/kg condition with saline. Following the suggestion by reviewer #2 we have now also improved the wording in the presentation of spectral results in the updated version of the manuscript to avoid potential misunderstanding:

“The focus of the EEG spectral analysis was on the comparison between the medium dose (5 mg/kg) of CNO and saline during NREM sleep because these two conditions were counterbalanced and performed first, excluding habituation effects to CNO or to the injection procedure. We observed that CNO injections were followed by a small but significant increase in spectral power in the range between 0.5 and 1.25 Hz and suppression of spectral power in nearly all frequency bins between 6 and 30 Hz during NREM sleep over the first two hours (Figure 3a and Figure 3 —figure supplement 2). While the increase in slow frequency bins during NREM sleep appeared to be temporary and does not persist after Benjamini-Hochberg correction for multiple testing (Benjamini and Hochberg 1995), the systematic suppression of power above 6 Hz was more robust and remained significant for both the acute (two hours) and the prolonged (six hours) observation period. Spectral analysis of wakefulness and REM sleep did not reveal any systematic effects of CNO (Figure 3 —figure supplement 2). A comparison of the other two CNO doses with saline injection indicated that 10 mg/kg CNO elicited similar effects on the NREM sleep spectrogram, but we found no systematic effects of the low CNO dose (1 mg/kg) on EEG spectra of any vigilance state (Figure 3 —figure supplements 1 and 3).”

– In line 194 you point to figure 1, but perhaps it is mean to point to figure 3a?

This error has been corrected.

– (methods) If I have it right either EEG or LFP were used for spectral analysis. This inconsistency likely adds to the variability.

We consistently used the frontal EEG derivation for spectral analysis. The mention of EEG/LFP in the methods section was included by mistake and has been removed.

– (methods) Comparisons between EEG ranges may require correction for multiple comparisons.

Please see our detailed response on the questions regarding the EEG spectral analysis above. We have now performed Benjamini-Hochberg correction for multiple comparisons and find a robust suppression of EEG power in the NREM spectrograms for frequencies above 6-8 Hz for the 5 and 10 mg/kg CNO conditions as well as for 3 mg/kg C21.

– (methods) Using 1 tailed t-test for the C21 experiments seems strange, although the logic is explicitly explained.

Experiments with small samples are likely to miss biologically relevant effects due to the lack in statistical power (Makin and Orban de Xivry 2019). One way to overcome this issue is to apply a one-sided t-test, when a unidirectional hypothesis can be clearly formulated. Based on the reported effects of CNO on specific sleep parameters in the control group of a previous study (Varin, Luppi, and Fort 2018) and based on our own CNO pilot experiments as well as based on previous work on sleep-modulation through clozapine in rats (Sorge, Pollmächer, and Lancel 2004), we had a clear hypothesis in which direction C21 effects would be expected. In order not to miss such effects despite the small sample size, we decided to apply one-tailed t-tests. While the usefulness of one-tailed t-tests (and in fact of null-hypothesis testing in general) should be debated in the scientific community, we provide effect size measurements that make the results comparable between experiments independent of the outcome of null-hypothesis testing (Makin and Orban de Xivry 2019). The effect size measurements, which are independent of the result of the null-hypothesis testing, indicate moderate-to-strong effects for several analyses and very strong effects for some sleep architectural parameters. The pattern of effects is also strikingly similar to the CNO experiment. We now provide the raw data and the statistical analysis tables in the online resources, which allow the readers to apply different statistical tests to the existing dataset and to further explore the data to substantiate the robustness of the findings. For the benefit of the reviewers, we have performed two-tailed t-tests for the C21 experiments and find that despite the considerable lack of statistical power 6 of the initially 9 significant sleep parameters remain significant. Only the REM latency, average REM episode duration, and the number of brief awakenings approach (p < 0.1), but do not reach the significance level of *p* = 0.05 when applying two-tailed instead of one-tailed t-tests.

– (methods) It seems that not all excluded mice are accounted for.

We have double-checked the methods section and find that all mice are accounted for. We now provide a comprehensive data table in the online materials which indicates the 5 animals with incomplete datasets and the reasons why individual recordings were not performed or not usable. As we show above for the benefit of the reviewers by analysing only the 11 animals that received all CNO conditions, our results are robust to the exclusion of the 4 animals that did not receive all conditions due to limits on the project license, and the 1 animal for which two recordings had to be omitted due to technical issues (electrical noise in one instance, considerable backflow after i.p. injection in the other instance).

Once again, we would like to thank the reviewers for their excellent suggestions for additional analyses, which have considerably strengthened the robustness of our findings and unequivocally support our conclusion that CNO and C21 modulate sleep in DREADD-free animals. Changes in and new additions to the main manuscript are highlighted in green and yellow in the Word document. The online repository is currently available under the private links provided below and will be published with a DOI upon acceptance of this manuscript.

Figshare online repository:

Summary table with information about the experiment: https://doi.org/10.6084/m9.figshare.21507561Individual recordings: https://doi.org/10.6084/m9.figshare.21507567Matlab analysis scripts: https://doi.org/10.6084/m9.figshare.21507573Matlab functions: https://doi.org/10.6084/m9.figshare.21507576GraphPad Prism files with source data, analysis sheets, and individual figures: https://doi.org/10.6084/m9.figshare.21507570

References

Achermann, Peter, and Alexander A. Borbély. 1998. “Coherence Analysis of the Human Sleep Electroencephalogram.” *Neuroscience* 85 (4): 1195–1208. https://doi.org/10.1016/s0306-4522(97)00692-1.

Adamantidis, Antoine, Carolina Gutierrez Herrera, and Thomas C Gent. 2019. “Oscillating Circuitries in the Sleeping Brain.” *Nature Reviews Neuroscience* 20 (12): 746–62. https://doi.org/10.1038/s41583-019-0223-4.

Banks, Gareth T, Mathilde C C Guillaumin, Ines Heise, Petrina Lau, Minghui Yin, Nora Bourbia, Carlos Aguilar, Michael R Bowl, Chris Esapa, Laurence Brown, Sibah Hasan, Erica Tagliatti, Elizabeth Nicholson, Rasneer Sonia Bains, Sara Wells, Vladyslav V. Vyazovskiy, Kirill Volynski, Stuart N Peirson, and Patrick M Nolan. 2020. “Forward Genetics Identifies a Novel Sleep Mutant with Sleep State Inertia and REM Sleep Deficits.” *Science Advances* 6 (33): eabb3567–eabb3567. https://doi.org/10.1126/sciadv.abb3567.

Benjamini, Yoav, and Yosef Hochberg. 1995. “Controlling the False Discovery Rate: A Practical and Powerful Approach to Multiple Testing.” *Journal of the Royal Statistical Society: Series B (Methodological)* 57 (1): 289–300. https://doi.org/https://doi.org/10.1111/j.2517-6161.1995.tb02031.x.

Ferrari, Loris L, Oghomwen E Ogbeide-Latario, Heinrich S Gompf, and Christelle Anaclet. 2022. “Validation of DREADD Agonists and Administration Route in a Murine Model of Sleep Enhancement.” *Journal of Neuroscience Methods* 380: 109679. https://doi.org/https://doi.org/10.1016/j.jneumeth.2022.109679.

Fleury Curado, Thomaz A, Huy Pho, Olga Dergacheva, Slava Berger, Rachel Lee, Carla Freire, Aya Asherov, Luis U Sennes, David Mendelowitz, Alan R Schwartz, and Vsevolod Y Polotsky. 2018. “Silencing of Hypoglossal Motoneurons Leads to Sleep Disordered Breathing in Lean Mice.” *Frontiers in Neurology*. https://www.frontiersin.org/article/10.3389/fneur.2018.00962.

Franken, P, A Malafosse, and M Tafti. 1999. “Genetic Determinants of Sleep Regulation in Inbred Mice.” *Sleep* 22 (2): 155–69. https://www.scopus.com/inward/record.uri?eid=2s2.0-0033559893&partnerID=40&md5=d97d8c0915f7e84511a76874fcfab91b.

Funk, Chadd M., Kayla Peelman, Michele Bellesi, William Marshall, Chiara Cirelli, and Giulio Tononi. 2017. “Role of Somatostatin-Positive Cortical Interneurons in the Generation of Sleep Slow Waves.” *The Journal of Neuroscience* 37 (38): 9132–48. https://doi.org/10.1523/JNEUROSCI.1303-17.2017.

Gent, Thomas C, Mojtaba Bandarabadi, Carolina Gutierrez Herrera, and Antoine Adamantidis. 2018. “Thalamic Dual Control of Sleep and Wakefulness.” *Nature Neuroscience* 21 (7): 974–84. https://doi.org/10.1038/s41593-018-0164-7.

Harding, Edward C, Xiao Yu, Andawei Miao, Nathanael Andrews, Ying Ma, Zhiwen Ye, Leda Lignos, Giulia Miracca, Wei Ba, Raquel Yustos, Alexei L Vyssotski, William Wisden, and Nicholas P Franks. 2018. “A Neuronal Hub Binding Sleep Initiation and Body Cooling in Response to a Warm External Stimulus.” *Current Biology* 28 (14): 2263-2273.e4. https://doi.org/https://doi.org/10.1016/j.cub.2018.05.054.

Hayashi, Yu, Mitsuaki Kashiwagi, Kosuke Yasuda, Reiko Ando, Mika Kanuka, Kazuya Sakai, and Shigeyoshi Itohara. 2015. “Cells of a Common Developmental Origin Regulate REM/Non-REM Sleep and Wakefulness in Mice.” *Science* 350 (6263): 957 LP – 961. http://science.sciencemag.org/content/350/6263/957.abstract.

Hubbard, Jeffrey, Thomas C Gent, Marieke M B Hoekstra, Yann Emmenegger, Valerie Mongrain, Hans-Peter Landolt, Antoine Adamantidis, and Paul Franken. 2020. “Rapid Fast-Δ Decay Following Prolonged Wakefulness Marks a Phase of Wake-Inertia in NREM Sleep.” *Nature Communications* 11 (1): 3130. https://doi.org/10.1038/s41467020-16915-0.

Huber, Reto, Tom Deboer, and Irene Tobler. 1999. “Prion Protein: A Role in Sleep Regulation?” *Journal of Sleep Research* 8 (SUPPL. 1): 30–36. https://doi.org/10.1046/j.1365-2869.1999.00006.x.

———. 2000. “Effects of Sleep Deprivation on Sleep and Sleep EEG in Three Mouse Strains: Empirical Data and Simulations.” *Brain Research* 857 (1–2): 8–19. https://doi.org/10.1016/S0006-8993(99)02248-9.

Jafari, Mohieddin, and Naser Ansari-Pour. 2019. “Why, When and How to Adjust Your P Values?” *Cell Journal* 20 (4): 604–7. https://doi.org/10.22074/cellj.2019.5992.

Kashiwagi, Mitsuaki, Mika Kanuka, Chika Tatsuzawa, Hitomi Suzuki, Miho Morita, Kaeko Tanaka, Taizo Kawano, Jay W Shin, Harukazu Suzuki, Shigeyoshi Itohara, Masashi Yanagisawa, and Yu Hayashi. 2020. “Widely Distributed Neurotensinergic Neurons in the Brainstem Regulate NREM Sleep in Mice.” *Current Biology* 30 (6): 1002-1010.e4. https://doi.org/https://doi.org/10.1016/j.cub.2020.01.047.

Kopp, Caroline, Jean-Marie Petit, Pierre Magistretti, Alexander A Borbély, and Irene Tobler. 2002. “Comparison of the Effects of Modafinil and Sleep Deprivation on Sleep and Cortical EEG Spectra in Mice.” *Neuropharmacology* 43 (1): 110–18. https://doi.org/10.1016/s0028-3908(02)00070-9.

Krone, Lukas B, Tomoko Yamagata, Cristina Blanco-Duque, Mathilde C C Guillaumin, Martin C Kahn, Vincent van der Vinne, Laura E McKillop, Shu K E Tam, Stuart N Peirson, Colin J Akerman, Anna Hoerder-Suabedissen, Zoltán Molnár, and Vladyslav V. Vyazovskiy. 2021. “A Role for the Cortex in Sleep–Wake Regulation.” *Nature Neuroscience* 24 (9): 1210–15. https://doi.org/10.1038/s41593-021-00894-6.

Makin, Tamar R, and Jean-Jacques Orban de Xivry. 2019. “Ten Common Statistical Mistakes to Watch out for When Writing or Reviewing a Manuscript.” Edited by Peter Rodgers, Nick Parsons, and Nick Holmes. *ELife* 8: e48175. https://doi.org/10.7554/*eLife*.48175.

McKillop, Laura E, Simon P Fisher, Linus Milinski, Lukas B Krone, and Vladyslav V. Vyazovskiy. 2021. “Diazepam Effects on Local Cortical Neural Activity during Sleep in Mice.” *Biochemical Pharmacology*, March, 114515. https://doi.org/10.1016/j.bcp.2021.114515.

McShane, B B, R J Galante, S T Jensen, N Naidoo, A I Pack, and A Wyner. 2010. “Characterization of the Bout Durations of Sleep and Wakefulness.” *Journal of Neuroscience Methods* 193 (2): 321–33. https://doi.org/10.1016/j.jneumeth.2010.08.024.

Singer, Michele Lynn, Sabhya Rana, Ethan S Benevides, Brian E Barral, Barry J Byrne, and David D Fuller. 2022. “Chemogenetic Activation of Hypoglossal Motoneurons in a Mouse Model of Pompe Disease.” *Journal of Neurophysiology*, August. https://doi.org/10.1152/jn.00026.2022.

Sorge, Silke, Thomas Pollmächer, and Marike Lancel. 2004. “Clozapine Alters Sleep-Wake Behavior in Rats.” *Neuropsychopharmacology* 29 (8): 1462–69. https://doi.org/10.1038/sj.npp.1300445.

Spierings, E L, M R Dzoljic, and M Godschalk. 1977. “Effect of Clozapine on the Sleep Pattern in the Rat.” *Pharmacology* 15 (6): 551–56. https://doi.org/10.1159/000136734.

Thomas, Christopher W, Cristina Blanco-Duque, Benjamin J Bréant, Guy M Goodwin, Trevor Sharp, David M Bannerman, and Vladyslav V Vyazovskiy. 2022. “Psilocin Acutely Alters Sleep-Wake Architecture and Cortical Brain Activity in Laboratory Mice.” *Translational Psychiatry* 12 (1): 77. https://doi.org/10.1038/s41398-022-018469.

Trachsel, Lorenz, Irene Tobler, Peter Achermann, and Alexander A Borbély. 1991. “Sleep Continuity and the REM-NonREM Cycle in the Rat under Baseline Conditions and after Sleep Deprivation.” *Physiology & Behavior* 49 (3): 575–80. https://doi.org/10.1016/0031-9384(91)90283-T.

Varin, Christophe, Pierre-Hervé Luppi, and Patrice Fort. 2018. “Melanin-Concentrating Hormone-Expressing Neurons Adjust Slow-Wave Sleep Dynamics to Catalyze Paradoxical (REM) Sleep.” *Sleep* 41 (6). https://doi.org/10.1093/sleep/zsy068.

Venner, Anne, William D Todd, Jimmy Fraigne, Hannah Bowrey, Ada Eban-Rothschild, Satvinder Kaur, and Christelle Anaclet. 2019. “Newly Identified Sleep–Wake and Circadian Circuits as Potential Therapeutic Targets.” *Sleep* 42 (5). https://doi.org/10.1093/sleep/zsz023.

[Editors’ note: what follows is the authors’ response to the second round of review.]

Essential revisions:1) Please add ANOVA prior to pairwise testing – it may also prevent unnecessary pairwise testing that would reveal a few significant scattered spectral bins that do not reflect a systematic change that the authors detail in the remainder of their response which focuses on multiple testing corrections for pairwise tests and bin size considerations.

We have now performed ANOVAs for the EEG spectral analysis and report the results in the main manuscript. The ANOVA results underscore the robustness of our previous findings and confirm that medium and high CNO doses (5 and 10 mg/kg CNO) as well as 3 mg/kg C21 systematically alter the NREM sleep spectrogram.

As expected, for those spectrograms where we had found only a few significant scattered spectral bins that do not reflect a systematic change, most two-way ANOVAs (‘frequency’, ‘condition’) did not result in significant main effects for ‘condition’ or for the interaction between ‘frequency’ and ‘condition’. More specifically, we found no main effects for ‘condition’ or interaction effects between ‘frequency’ and ‘condition’ in any of the wake spectra. We also found no main effects for ‘condition’ or interaction effects between ‘frequency’ and ‘condition’ in the NREM sleep spectra of the 1 mg/kg condition. However, the REM sleep spectral analysis indicated an interaction effect for ‘frequency’ and ‘condition’ without a main effect for ‘condition’ for the 6-hour time window in the 1 mg/kg CNO condition as well as in the 2- and 6-hour time windows in the 10 mg/kg CNO condition.

Most importantly, and consistent with our previous results, the ANOVAs for the NREM sleep spectrograms of the 5 and 10 mg/kg CNO condition compared to saline, where we had found several neighbouring spectral bins with significant differences in the uncorrected paired t-tests for individual bins, revealed a significant main effect for ‘condition’ and an interaction effect between ‘frequency’ and ‘condition’.

For the NREM spectrograms in the C21 condition, where we had also found several neighbouring spectral bins with significant differences in the uncorrected paired t-tests for individual bins, the ANOVAs reveal a significant effect for the interaction between ‘condition’ and ‘frequency’ in the 2- and 6-hour time windows and a significant main effect for ‘condition’ in the 6-hour time window. In the 2-hour time window the main effect for ‘condition’ approached but did not reach the significance threshold of p = 0.05 (F_(1,5)_ = 6.560, p = 0.0506).

We have removed all results of pairwise testing from those spectrograms where two-way ANOVAs did not reveal an interaction effect between ‘frequency’ and ‘condition’. We report the ANOVA results in the updated manuscript, except for the main effects of ‘frequency' that were – of course – significant in all spectral analyses. For the CNO conditions the ANOVA results are listed in the new supplementary table 4. For the C21 condition, the results are presented in the figure caption of Figure 5 —figure supplement 1.

For the presentation of the absolute EEG spectra in the supplementary data (Figure 3 —figure supplements 1-3 and Figure 5 —figure supplement 1a), we believe the most informative illustration of post-hoc tests following a significant interaction effect of ‘frequency’ and ‘condition’ is to highlight those frequency bins that reach the α error threshold of p=0.05 in uncorrected paired t-tests (grey asterisks), those that reach the α error threshold of p=0.01 in uncorrected paired t-tests (black asterisks), and those frequency bins that are significant following Benjamini-Hochberg correction for multiple testing (blue asterisks). To simplify the figures, we have now removed the dark blue asterisks that we had added as a fourth line in the previous revision illustrating that reducing the bin size resolution from 0.25 to 1 Hz before Benjamini-Hochberg correction does not alter the results.

We also include the ANOVA results for the NREM sleep spectral analysis in the C21 vs. saline comparison:

2-hour time window:

Main effect ‘condition’: *F*_(1,5)_ = 6.560, *p* = 0.0506Interaction ‘condition’ x ‘frequency’: *F*_(118,590)_=2.961, *p* < 0.0001 6-hour time window:Main effect ‘condition’: *F*_(1,5)_ = 10.32, *p* = 0.0237Interaction ‘condition’ x ‘frequency’: *F*_(118, 590)_ = 4.726, *p* < 0.0001

2) Related, the authors refer to the pairwise testing of EEG spectral data as "post-hoc" testing, but statistically speaking, none of these comparisons are "post-hoc" because ANOVAs were not conducted prior to pairwise testing. See Figure 3, Figure 5, Figure 3 —figure supplement 1, Figure 3 —figure supplement 2, Figure 3 —figure supplement 3, and Figure 5 —figure supplement 1.

We have now conducted ANOVAs prior to pairwise testing for the EEG spectral data. We have adjusted the presentation of the EEG spectral data as outlined above and updated the wording in the figure legends. We have also updated the relevant paragraph in the methods section.

Please consider additional points for discussion as suggested by the reviewers.Reviewer #1 (Recommendations for the authors):The authors did a comprehensive job in addressing the concerns raised in the first round of review. The manuscript improved significantly. I only have a few final suggestions, which are all minor; the authors should decide on whether and how to incorporate them into the final manuscript:1. In their response, the authors write that "While we agree with the reviewer that the effect of CNO and C21 on the overall amount of sleep could be considered modest – and therefore sleep-modulating effects of these actuators have often been missed in the past – our results that … are strong and robust."I think that the authors may consider conveying this message also to the readers e.g. in Discussion. The fact that some effects can be considered modest, yet consistent and of potential important physiological significance, may be important to explicitly acknowledge in terms of relation to previous studies and as something for future chemogenetic studies for all the community.

We thank the reviewer for the suggestion to include this discussion point from our previous point-to-point response in the main manuscript. We have now amended the Discussion section stating:

“We speculate that the sleep modulating effects of CNO and C21 have been overlooked in the past, because the change in the overall amount of sleep, and in the proportion of time spent in the respective vigilance states, is modest, and many studies only analysed these measures in their DREADD-free control groups. However, the effects of both DREADD actuators on the relative amount of REM sleep, sleep architecture, EEG power spectra of NREM sleep, sleep stability and continuity are strong, robust and highly relevant for future chemogenetic studies.”

2. It could be worthwhile to separately show the effects during the first 2h in Figure 2. Most people/studies employing chemogenetics only look at first few hours and this may be most relevant for them.

This is a logical suggestion and we have looked at sleep architecture during the first 2 hours. Due to the suppression of REM sleep, some animals have no and others only one, two or three REM sleep episodes following CNO/C21 injections. Therefore, the analysis of REM architecture over this short time frame in which REM sleep is suppressed is not meaningful.

However, all animals have sufficient NREM sleep in the first 2 hours for a robust analysis of NREM sleep architecture. We have included these new analyses as supplementary tables 3 and 6 and we include Author response image 6 and Author response image 7 for the benefit of the reviewers and editors, which we would be happy to add as figure supplements to Figures 2 and 5 at the discretion of the editors. The effects closely resemble the findings for the 6-h observation time window.

**Author response image 6. sa2fig6:** NREM sleep architecture over the first 2 hours following saline and CNO injections.

**Author response image 7. sa2fig7:** NREM sleep architecture over the first 2 hours following saline and C21 injections.

3. It could be worthwhile that limitations section in Discussion includes that C21 only had one dose (3 mg/kg) rather than a more complete dose-response assessment (judging by the CNO data, effects may vary depending on dose so this may be important)

This is an excellent discussion point which we have now included in the limitations section of the discussion in the main manuscript:

“It should also be highlighted that we only included a single dose of C21 (3 mg/kg) and therefore a dose-response assessment should be performed next to assess a putative dose dependency of C21 effects on sleep.”

4. In Discussion lines 420-> the authors suggest that "it is vital to minimize the injected doses of DREADD actuators" but sometimes lower or medium doses have stronger effects, so this statement may be overly simplistic

We agree with the reviewer that this statement was overly simplistic. Indeed, our data shows that the effects of DREADD actuators are mostly, but not always dose-dependent. In addition, the dose of DREADD actuators needed to elicit a DREADD-mediated effect can vary between the type of experiment (e.g. inhibitory DREADDs generally need higher doses to be activated than excitatory DREADDs) and between the route of application, the galenic formulation of the actuator, the expression levels of the DREADD receptors, etc.

We have reworded the sentence as follows to provide a more nuanced statement:

“…our study supports the proposal to use the lowest dose of a DREADD actuator sufficient to elicit a DREADD-mediated effect in a respective experiment and to include a non-DREADDexpressing control group injected with the same dose of the respective actuator as the DREADD-expressing group in each individual experiment.”

Reviewer #2 (Recommendations for the authors):This resubmission by Traut, Mengual et al. is much improved. The additions and edits now provide a more complete and accurate presentation of the data especially for REMS time and architecture.Including REMS time as a % of total recording time (TRT) now provides a more informed understanding of REMS as % of total sleep time (TST) because it provides context for the TST metric (i.e., is NREMS changing, is REMS changing, or are both states changing). Reporting both metrics is how other groups the authors cited (Franken, Malafosse, and Tafti 1999; Huber, Deboer, and Tobler 1999; Kashiwagi et al. 2020) approached the analysis as well. We can now see that the impact of CNO on REMS is nuanced preserving overall REMS amounts but reducing REMS relative to NREMS for higher doses of CNO. It might be helpful to include the REMS as a % of TRT data as the third row in Figures 1 and 4 (after NREMS and before REMS as a % of TST) as the papers cited above did. The reason being that, looking at the data altogether, one could reasonably posit that it is actually increasing NREMS more so than decreasing REMS driving the reduction in the REMS % TST value after higher doses of CNO at least for the acute phase. But that could be left to the interpretation of the reader.

We thank the reviewer for the positive assessment of our revision.

We added the analysis “REMS as a % of TST” to Figures 1 and 4 as suggested and adjusted the figure supplements of Figures 1 and 4 accordingly to avoid duplication of the figure panels.

Overall, the authors adequately addressed the majority of my concerns, but a few remain as detailed below:1) I appreciate the authors detailed response for multiple testing considerations and for now including those analyses. However, the rationale for not conducting an ANOVA for the EEG spectral data prior to (what is typically post-hoc) pairwise testing is lacking. The authors cite Achermann and Borbély (1998) in their response to support not conducting an ANOVA; however, Achermann and Borbély state that they conducted pairwise testing post-hoc only if the ANOVA for repeated measures was significant. This testing occurred over 198 frequency bins. Using an ANOVA prior to pairwise testing may also prevent unnecessary pairwise testing that would reveal a few significant scattered spectral bins that do not reflect a systematic change that the authors detail in the remainder of their response which focuses on multiple testing corrections for pairwise tests and bin size considerations.

We appreciate the reviewer’s interest in assessing whether the systematic changes in EEG spectra can be supported with significant ANOVA results and whether in turn the ANOVA results are not significant for those spectra where we only found a few scattered spectral bins. We have now conducted the ANOVAs as outlined at the beginning of this response and find our previous results confirmed.

We apologise that our citation of Achermann and Borbély (1998), where the authors argue against corrections for multiple testing on EEG spectral analysis “since adjacent frequency bins do not vary independently”, was misplaced and we agree with the reviewer that ANOVAs represent a suitable statistical approach to test the robustness of systematic EEG spectral changes observed in pairwise testing of individual frequency bins.

2) Related, the authors refer to the pairwise testing of EEG spectral data as "post-hoc" testing, but statistically speaking, none of these comparisons are "post-hoc" because ANOVAs were not conducted prior to pairwise testing. See Figure 3, Figure 5, Figure 3 —figure supplement 1, Figure 3 —figure supplement 2, Figure 3 —figure supplement 3, and Figure 5 —figure supplement 1.

We thank the reviewer for pointing this out and refer to our response above.

3) To improve clarity, it would be helpful to edit Page 6, Line 125 and Page 20, Line 345 to "suppression of REM sleep relative to NREM sleep." Similarly, editing Figure 3e from "REM sleep" to "REM sleep (%TST)" or REM/NREM sleep" would also help.

We have implemented the suggested edits to improve the clarity of the data presentation in the updated manuscript. We would like to clarify, however, that while we only find a relative suppression of REM sleep (REM as %TST and in the REM/NREM ratio) following CNO injections, Varin et al. (2018) show an absolute suppression (REM as % of recording time) of REM sleep in their supplementary data and we also find an absolute suppression of REM in the C21 condition. In our view, the most nuanced approach to avoid black-white interpretation of null-hypothesis testing is to look at the effect sizes, which indicate a moderate reduction in the absolute amount of REM and a stronger reduction in the REM/NREM ratio. *4*

4) On Page 12, Lines 215 – 216, the authors state that less REM sleep is typically observed during recovery from sleep deprivation. This is not typically the case, and the cited supporting text (Huber, Deboer, and Tobler 2000) shows that REMS either increases or does not differ after sleep deprivation compared to baseline.

We thank the reviewer for pointing out that our wording was easily misunderstood. The reviewer is correct that Huber et al. showed that the total amount of REM sleep does not differ or is mildly increased following sleep deprivation compared to baseline. However, NREM sleep is strongly increased. This means that sleep deprivation primarily leads to a considerable NREM rebound, which in turn can induce a reduction in the relative amount of REM sleep. However, this is not consistently observed across mouse strains and varies between the time of day when sleep deprivation is performed as well as between the early and late recovery phase. We have removed this statement from the manuscript to avoid further misunderstanding.

5) The authors state in the Figure 1 —figure supplement 1 caption, "Note the relatively high variability in the percentage of REM sleep as percentage of total recording time between animals (bottom row panels)…" (Page 36, Lines 704 – 705). This statement is not supported by the data shown. The individual variability and error bars are similar across all analyses displayed in these plots. Also, the SEM values listed in Supplementary Table 1 are lower in all conditions calculated as %TRT compared to %TST, and the SEM values are of similar magnitude relative to their respective means for both data sets. This also applies to the statement on Page 8, Line 133 about high variability of sleep amounts between individual animals.

The reviewer might have overlooked that the scaling of the y-axis is different in the presentations of REM sleep as percentage of total recording time (%TRT) and of total sleep time (%TST). To double-check that our statement was correct, we z-transformed the data and find that the standard deviation is systematically larger when REM sleep is calculated as %TST compared to when it is expressed as % TRT (CNO 1 mg/kg: 1.67 vs. 1.073; 5 mg/kg: 1.955 vs. 1.468; 10 mg/kg 1.291 vs. 1.142). Nevertheless, to simplify the presentation of the results and avoid any speculation in the Results section, we have removed the statements in line with the reviewer’s suggestion.

6) Figure 3 caption should say post-hoc comparisons are shown for panels c and d.

We’ve now corrected this mistake in the legend of Figure 3.